# Parallel G-quadruplexes recruit the HSV-1 transcription factor ICP4 to promote viral transcription in herpes virus-infected human cells

Ilaria Frasson [1], Paola Soldà[1], Matteo Nadai [1], Sara Lago [1] & Sara N. Richter [1✉]

G-quadruplexes (G4s) are four-stranded nucleic acid structures abundant at gene promoters. They can adopt several distinctive conformations. G4s have been shown to form in the herpes simplex virus-1 (HSV-1) genome during its viral cycle. Here by cross-linking/pull-down assay we identified ICP4, the major HSV-1 transcription factor, as the protein that most efficiently interacts with viral G4s during infection. ICP4 specific and direct binding and unfolding of parallel G4s, including those present in HSV-1 immediate early gene promoters, induced transcription in vitro and in infected cells. This mechanism was also exploited by ICP4 to promote its own transcription. Proximity ligation assay allowed visualization of G4-protein interaction at the single selected G4 in cells. G4 ligands inhibited ICP4 binding to G4s. Our results indicate the existence of a well-defined G4-viral protein network that regulates the productive HSV-1 cycle. They also point to G4s as elements that recruit transcription factors to activate transcription in cells.

[1] Department of Molecular Medicine, University of Padua, Padua, Italy. ✉email: sara.richter@unipd.it

DNA can adopt a variety of non-canonical structures that play important functional roles[1]. G-quadruplexes (G4s), secondary structures that form in guanine-rich regions, are among the most stable non-canonical structures[2]. G-quartets, the building blocks of G4s, are formed by an extensive hydrogen-bonding network that links four guanine bases around a cationic core. G4 structures comprise G-quartets stacked upon each other, stabilised by base stacking between the layers. G4s are highly polymorphic and can adopt different topologies (i.e. parallel, antiparallel or hybrid topology), based on strand stoichiometry and polarity, the nature and length of the loops and their location in the sequence[3]. At the cellular level, the parallel G4 conformation is the most widespread since it is adopted by RNA G4s[4] and by many DNA G4s in physiological conditions[5–8]. In addition, it is functionally recognised by G4 interacting antibodies and proteins[9–12].

G4 folded/unfolded state modulates numerous processes, including replication, transcription, translation and can trigger genomic instability[3,13–15]. G4-forming sequences are highly represented in the genomes of all living beings[16], indicating that these structures are likely exploited as regulatory elements, while their negative effects can be mostly avoided[2]. Characterisation of the G4-protein interactome in some cases allowed the deep understanding of the G4-mediated biology[17].

G4-ligands are molecules that selectivity target G4s by differentiating them from the double-stranded DNA; in principle they can also selectively target the specific G4, based on its structural uniqueness, and indeed some degree of selectivity has been reported by different strategies[18–21].

The human *herpesviruses* constitute a large family of dsDNA viruses, which are the causative agents of a wide range of diseases, spanning from oral/genital blisters to cancer. In particular, herpes simplex virus-1 (HSV-1) infects ~70% of the adult population, establishing a latent neuronal infection which, upon stressful conditions, may be reactivated to a lytic symptomatic state. HSV-1 is the major cause of blindness and viral encephalitis worldwide. HSV-1 life cycle is characterised by a coordinated and sequential cascade of expression of three temporal classes of viral genes: the viral tegument protein VP16 prompts transcription of the immediate early (IE) viral genes, the products of which in turn activate transcription of the early (E) and late (L) viral genes; transcription of the late viral genes is also coupled to viral DNA replication[22]. We have previously demonstrated that biologically significant and highly conserved G4s are present in crucial elements of the HSV-1 genome, in particular, within repeated regions and in immediate early gene promoters, which control the viral life cycle[23,24]. Viral G4s are observable in infected cells with peak signals during HSV-1 DNA replication[25].

Here we show that the viral transcription factor ICP4, which is expressed soon after infection and sustains the HSV-1 viral cycle, displays substantial binding preference for parallel G4s. This essential viral protein unfolds G4s in viral promoters and thereby regulates HSV-1 viral gene expression. We visualised ICP4/G4 complexes at the single G4 level in infected cells and showed that ICP4 modulates its own promoter via G4-mediated interaction. Our results indicate the existence of a well-defined G4-viral protein network and of previously unexplored viral strategies that regulate the productive HSV-1 cycle.

## Results

**HSV-1 transcription factor ICP4 binds to G4-folded nucleic acids.** We have recently demonstrated that the HSV-1 genome is significantly enriched in conserved G4 patterns and that the amount of G4s that form in HSV-1-infected cells depends on the viral step, being the highest during viral DNA replication[23,25]. To gain further insights into the G4-dependent virus biology, we set out to identify viral/cellular proteins that could specifically interact with biotinylated G4-forming HSV-1 sequences. We employed two sequences that have been previously reported to fold into G4s and be highly repeated within the HSV-1 genome: the 58 nt-long un2L2 and 66 nt-long gp054dL3 G4-folding oligonucleotides (Supplementary Table S1), which are highly conserved and repeated within the terminal and internal repeat short regions (TRS and IRS, RS2) and the UL36 coding sequence (CDS, UL RS1), respectively[23,26]. Un2L2 is composed of 9 G-tracts, ranging from three to nine G bases each, gp054dL3 of ten G-tracts of 4–8 Gs each. Two additional G4-forming sequences were used: the 28-nt-long LTR-III from the HIV-1 LTR promoter (6 G-tracts of 2–3 Gs)[27,28], and the 26 nt-long myc G4-forming region from the myc oncogene promoter (6 G-tracts of 2–4 Gs)[29,30] (Supplementary Table S1). These two sequences were used as positive controls, representing well-characterised viral and cellular G4s, respectively. Circular dichroism (CD) spectra were acquired to characterise the G4 conformation of the test sequences, since CD peaks represent signatures of the parallel (positive and negative peaks at 260 and 240 nm, respectively), antiparallel (positive and negative peaks at 290 and 260 nm, respectively) and hybrid or mixed topology (positive peaks at 290 and 260 nm)[31]. Biotinylated un2L2 and myc form parallel G4s, characterised by a main peak at 260 nm and no peak at 290 nm (signal ratio at 260/290 nm of 13.7 and 10.9, respectively; Supplementary Fig. S1); in contrast, biotinylated LTR-III and gp054dL3 display a hybrid/mixed conformation, characterised by CD peaks at both 260 and 290 nm (signal ratio 260/290 nm of 3.1 and 0.8, respectively). A G-rich sequence, unable to fold into G4, was used as control (Supplementary Table S1 and Supplementary Fig. S1). These oligonucleotides were next incubated with nuclear extracts from U-2 OS human osteosarcoma cells infected with HSV-1 strain F (Multiplicity of Infection, MOI of 2). U-2 OS cells are susceptible and permissive to HSV-1[32]. Nuclear extracts were collected at 8 h post infection (hpi); the HSV-1-derived oligonucleotide un2L2 was additionally analysed at 16 hpi. These two time points were chosen to analyse proteins present during viral DNA replication (8 hpi), i.e. proteins involved in transcription, DNA replication, chromatin remodelling and RNA processing, and at the final step of virion maturation (16 hpi), i.e. proteins involved in post-transcriptional RNA processing and viral structural proteins[22,33]. The proteins bound to the un2L2 bait were subjected to SDS–PAGE/mass spectrometry (MS): data were analysed by Mascot software, which assigns a score based on the number of fragments that match the recognised protein and the probability that the observed match is not a random event. Results are summarised in Supplementary Table S2: only proteins with score >30 in duplicate experiments and displaying selectivity towards the G4 structure, (i.e. score on the G4 at least 100 times higher than the score on the G-rich sequence unable to form G4) were retained. At 8 hpi, five viral (UL42, ICP4, DUT, UL23) and two cellular (HNRNPA1L2 and NCL) proteins, all involved in transcription regulation and replication, were retrieved. At 16 hpi, three additional viral proteins were recovered: UL18, UL19 and UL38, all of which are part of the viral encapsidation machinery. At this time post-infection, despite the relative increase in structural proteins including tegument proteins, UL42 and ICP4, two proteins recovered at 8 hpi and belonging to the replication and transcription machinery, retained the most efficient G4 binding. No additional proteins were recovered with the other G4 baits: ICP4, UL42, UL19 with gp054dL3; ICP4 with LTR-III; UL23, ICP4, UL42 and UL19 with myc (Supplementary Table S2). ICP4 was the only protein that interacted with the four G4 baits and, interestingly, its Mascott score was proportional to the G4 bait 260/290 nm ratio, i.e., with the degree of G4 parallel

conformation. To test the reliability of the pull-down/MS data we assessed binding to G4s of ICP8, a viral zinc-metalloprotein that was never detected in our pull-down/MS assay. We selected this protein because it has been reported to bind DNA during HSV-1 DNA replication[34], to localise and bind to telomeric DNA and to colocalize with an anti-G4 antibody[35,36]. We expressed and purified ICP8 from bacterial cells[37] and assessed its binding towards different G4 sequences in both sodium and potassium buffers via electrophoresis mobility shift assay (EMSA; Supplementary Fig. S2a). Bacteria-purified ICP8 was active and was able to bind the positive control sequence[38] but did not show any binding towards the tested G4 sequences (HSV-1 un2, HIV-1 LTR-III and the cellular hTel). To overcome the possibility that purification from bacteria modified ICP8 nucleic acid recognition, we set up a pull-down assay coupled to western blot (WB) from infected U-2 OS cells (Supplementary Fig. S2b). ICP8 expressed during the viral cycle displayed no binding affinity towards the G4s. We next checked whether proteins such as ICP0, Histones H1 and H3 that were previously proven not to be associated with HSV-1 DNA during replication[33,39] were present in the MS output, even at low level. None of these proteins was ever recovered with any G4s or control sequences. These data altogether confirmed the quality and reliability of the pull-down/MS assay. Taken together these data prove that we had reliable data from our pull-down assay coupled to MS analysis, with novel viral proteins identified as G4 binders at the HSV-1 genome level, with ICP4 being the most noteworthy.

**ICP4 specifically recognises and unfolds parallel G4s.** Among the newly identified viral G4-binding proteins, we selected ICP4 for further investigation, as this protein was recovered with all G4 baits and it is the most important HSV-1 transcription factor. To assess if ICP4 directly interacted with the tested G4s, we set up a pull-down assay coupled to western blot (WB) to detect only proteins directly interacting with the test oligonucleotides. ICP4 binding was assessed against several G4s: un2L2, its shorter sequence un2 and gp054a from the HSV-1 genome;[23] LTR-III and its longer sequence LTR-II + III + IV from the HIV-1 genome;[28] myc and VEGF from the human genome and BOM17 from Bombyx mori telomere region[30,40,41]. In addition, HIV-1 LTR-III i-motif forming sequence (LTR-IIIc) and the reported ICP4 consensus sequence (IE3) were used as non-G4 and positive controls, respectively[42,43]. The chosen G4 sequences differed in length (from 17 to 58 nt) and topology (parallel, antiparallel or hybrid, identified by 260/290 nm ratio in the range of −0.3 to 14.6, Fig. 1a, Supplementary Fig. S3a and Supplementary Table S1). The cross-linking-pull-down assay confirmed that ICP4 directly binds to the G4s (Fig. 1b and Supplementary Fig. S3b), with clear preference for G4s with high 260/290 nm ratio, thus those with a parallel topology, such as un2L2, myc and VEGF, that were bound with 2.5-fold higher affinity than the ICP4 consensus sequence IE3. In contrast, oligonucleotide length was not a determinant of recognition, as we observed negligible difference between un2 and un2L2 (29 vs 58 nts, respectively), and LTR-III and LTR-II + III + IV (28 vs 45 nts, respectively) sequences (Supplementary Fig. S3).

To assess ICP4 activity on G4s, we purified ICP4 from infected cells via immunoprecipitation (ipICP4) using a monoclonal anti-ICP4 antibody (Ab) or via ultrafiltration (ufICP4)[44]. In the first case we obtained low amounts of very pure protein, in the second larger amounts with acceptable purity. The purity and identity of the protein were confirmed by WB and mass spectrometry (MS) analysis (Supplementary Fig. S4). UfICP4 was recovered in its multiband native state, encompassing the monomer (175 KDa), the homodimer (heavier than 245 KDa) and the differentially

phosphorylated bands[45] (Supplementary Fig. S4a). No other proteins, especially those that are often reported to co-precipitate with ICP4, i.e., TBP (TATA BOX binding protein) and TFIID[46] were recovered, as assessed by MS evaluation (Fig. S4c). To check that ufICP4 retained its G4 binding capacity upon purification we set up an EMSA assay using un2 and the G-rich-scrambled oligonucleotides as positive and negative control, respectively (Supplementary Fig. S4d). Bands corresponding to ICP4-oligonucleotide complexes were obtained exclusively with the folded oligonucleotide, as reported[47]. In the case of ipICP4, the use of the anti-ICP4 antibody allowed the immunoprecipitation of the monomeric ICP4 (135 KDa) towards which the antibody has the highest affinity (Supplementary Fig. S4b). In our hands the ultrafiltration procedure consented higher protein yields with a negligible cost (15-fold cheaper than immunoprecipitation) and was, thus, chosen as the main purification technique to obtain functional fractions of ICP4 for subsequent experiments. Both ipICP4 and ufICP4 were used to assess their activity on fluorescence labelled G4 oligonucleotides by fluorescence resonance energy transfer (FRET) analysis. As the main determinant of the FRET signal is the distance between the donor (FAM) and acceptor (TAMRA) fluorophores, when these are linked to the 5′ and 3′-end of a G4-forming oligonucleotide, the intensity of the FRET signal is directly proportional to G4 folding. We first ascertain that the presence of the FAM and TAMRA fluorophores at the 5′- and 3′-end, respectively, of the test G4s, i.e. two viral (un2 and LTR-III) and two cellular (myc and hTel) G4s, induced only minor modifications on their overall G4 topology and melting temperature, as proven by CD analysis (Supplementary Fig. S5a and Supplementary Table S1): un2 and myc maintained their parallel (260/290 value of 25.0 and 52.2, respectively), and LTR-III and hTel their hybrid G4 conformation (260/290 value of 1.2 and 0.3, respectively). The fluorescence signal and its variation upon incubation with ICP4 allowed to calculate the energy transfer (E) and end-to-end distance (R) between the two fluorophores, and therefore the degree of G4 folding. In the case of un2 and myc, addition of ipICP4 induced unfolding (Fig. 1c and Supplementary Fig. S5c–f), the degree of which largely or mildly increased when the G4 complementary strand was added before or concurrently to ipICP4, respectively (Supplementary Table S3 and Supplementary Fig. S5b–g). Addition of the complementary strand alone induced only minor unfolding, while the control BSA protein did not show any effects. Unfolding was more pronounced on myc than un2 likely because of the lower initial stability of myc that forms a three-tetrad G4, vs un2, which is a four tetrad G4[48] (Supplementary Table S3). Similar unfolding with increased efficiency was obtained when ufICP4 was used (Supplementary Table S3 and Supplementary Fig. S5f–g). The improved activity was likely due to the milder purification conditions of ufICP4 vs ipICP4. UfICP4-mediated unfolding pattern was confirmed also when monitoring the FRET signal at 37 °C (105 and 103% of unfolding on un2 and myc, respectively, Supplementary Fig. S5h). When BRACO-19 (B19), a documented stabilising G4 ligand[49], was added, we observed stabilisation of un2. When ufICP4 was added concurrently to B19, the unfolding degree was much lower than that induced by ufICP4 alone, indicating that B19 was able to partly inhibit ICP4 unfolding activity (Supplementary Table S3). In contrast, negligible ICP4-mediated unfolding was observed on the hybrid G4s, i.e., LTR-III and hTel (Supplementary Table S3 and Supplementary Fig. S5d–e).

CD was next employed to further investigate ICP4 unfolding activity. UfICP4 was used because CD requires much higher protein amounts than FRET. First, we measured CD spectra of the unlabelled oligonucleotides (Supplementary Fig. S6): we noticed that the unlabelled un2 folded into antiparallel G4, in line

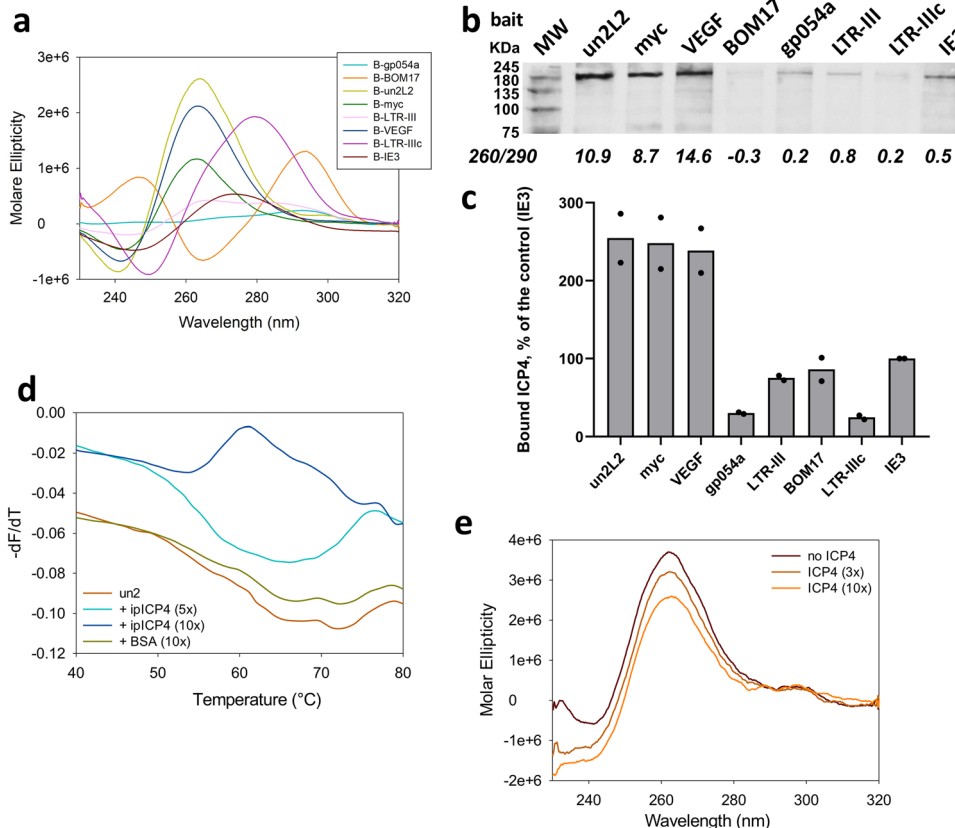

**Fig. 1 ICP4 binding and unfolding of G4-folded oligonucleotides. a** CD spectra of the biotinylated oligonucleotides used in the cross-linking-pull-down assay, folded in potassium phosphate buffer (20 mM PB, 80 mM KCl). Spectra were acquired from two independent experiments with two replicates each; spectra from one measurement are shown. **b** Western blot analysis of ICP4 binding to the indicated G4s (parallel: un2L2, myc and VEGF; antiparallel BOM17; mixed gp054a and LTR-III; non G4 LTR-IIIc; ICP4 consensus binding sequence: IE3). Infected cell nuclear extracts (5 µg) were incubated with formaldehyde-activated G4-folded oligonucleotides; proteins bound to the G4s were eluted after high stringency washes. ICP4 was detected using the anti-ICP4 antibody. MW molecular weight (Marker VI, Applichem). Below the gel image, the 260/290 ratio for each biotinylated oligonucleotide is displayed. **c** Quantification of ICP4 bound to the G4s in (b) is indicated in the bar chart (average of two independent experiments, each oligo was tested in one replicate per experiment, SD is displayed). Band quantification was normalised for the fraction of ICP4 bound to its consensus sequence (IE3). **d** First derivative FRET-melting curves ($-dF_{525}/dT$ vs T) of F-un2-T G4 oligonucleotide treated with ICP4 or BSA at the indicated protein/DNA ratios, in 20 mM PB pH 7.4, 2 mM KCl. Spectra were acquired from two independent experiments with two replicates each; spectra from one measurement are shown. **e** CD spectra showing the unfolding (molar ellipticity reduction) of un2L2 G4 in the presence of increasing ufICP4 concentrations. The protein spectrum was subtracted from DNA-protein complex spectra. Two independent experiments were performed, with one replicate per condition.

with previously reported evidence that end modifications may modify the overall G4 topology[50]; in contrast, its longer sequence, un2L2, maintained the parallel G4 conformation. The unlabelled myc and LTR-III/hTel also maintained the parallel and mixed conformation, respectively, observed with the labelled oligonucleotides. ICP4 decreased the CD signal, i.e., unfolded, of the parallel un2L2 and myc G4s, while had no effect on the antiparallel un2 (Fig. 1d and Supplementary Fig. S6), data that clearly confirm the G4 topology- rather than G4 sequence-dependence of ICP4 activity. In addition, the CD signal of LTR-III, reported to fold into a 3 + 1 hybrid G4[27], was decreased to a negligible extent, while only the signal corresponding to the parallel topology was affected in hTel, which adopts several different conformations in solutions[51] (Supplementary Fig. S6). The decrease in molar ellipticity was proportional to the 260/290 value: higher with 260/290 value >10 (parallel G4s); lower with 260/290 value >0 and <1 (hybrid G4s); null for 260/290 value <0 (antiparallel G4s).

To confirm ICP4 unfolding ability, we investigated whether ICP4 could improve the rate and speed of duplex formation of a G4 folded oligonucleotide in the presence of its complementary strand. The G4-folded F-un2-T oligonucleotide (Supplementary Table S1) was incubated for 15, 60 and 120 min with its complementary strand in the presence or absence of ufICP4. Addition of the complementary strand induced formation of the duplex in a time dependent manner; the duplex migrated slower than the G4-folded oligonucleotide in EMSA (Supplementary Fig. S7a, b). In the presence of ufCP4, the duplex formed more and faster with respect to the reactions with the complementary strand alone (Supplementary Fig. S7c). The un2 duplex was characterised by two bands with similar electrophoretic mobility, as already reported for other G4s[52]. The two bands likely correspond to two duplex species with different conformation or a different degree of complementarity. The free G4 species did not completely disappear due to the extremely high stability of un2 G4 in solution.

These in vitro data indicate that ICP4 specifically binds and unfolds parallel G4s, favouring duplex formation.

**ICP4-G4 interaction takes place in the nucleus of infected cells during HSV-1 infection.** To verify ICP4 interaction with HSV-1 G4s in the infected cells, we first investigated the actual G4 folding of the un2 sequence in HSV-1 infected U-2 OS cells by FISH analysis with a biotinylated DNA probe that does not fold

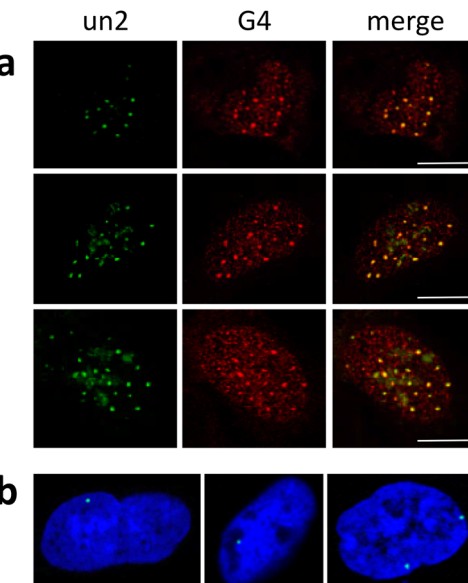

**Fig. 2 ICP4 and un2 G4 colocalization in the nucleus of HSV-1 infected cells. a** Representative images of immuno-DNA-FISH in U-2 OS cells infected with HSV-1, strain F, MOI of 3, fixed at 8 hpi. The green and red signals correspond to un2 G4 and anti-G4 Ab signals, respectively; the merged signals are in yellow. Scale bars 10 μm. Controls are shown in Supplementary Fig. S5; **b** Representative images of proximity ligation assay (PLA) in U-2 OS cells infected with HSV-1, strain F, MOI of 3, fixed at 8 hpi. The green dots (PLA signal) correspond to G4-ICP4 interaction (one or two dots per nucleus). The blue signal corresponds to TOTO3 nuclear staining. Scale bars 10 μm. Controls are shown in Supplementary Fig. S6. Both Immuno-DNA-FISH and PLA experiments were performed in three independent assays per tested condition.

into G4 and partially overlaps with the un2 sequence without hindering its G4 folding; we visualised G4s with an anti-G4 Ab[53] (Supplementary Table S1). The majority of the un2 signal co-localised with the G4 Ab (Fig. 2a and Supplementary Fig. S8), indicating that un2 is indeed folded into G4 in cells in these conditions. We next investigated ICP4 and un2 G4 colocalization by immuno-staining/immuno-DNA-FISH in U-2 OS cells infected with HSV-1 at MOI 3, that, according to the Poisson's distribution, allows infection of up to ~95% cells, at different times post infection (2–8 hpi). We used a biotinylated DNA probe partially overlapping with un2 (Supplementary Fig. S9a) and an anti-ICP4 monoclonal antibody. We observed one or two over-lapping spots per nucleus up to 4 hpi, while multiple merged dots were visible at 6 and 8 hpi (Supplementary Fig. S9b). At the higher time points, however, ICP4 signal was so widely dis-tributed within the nucleus[54,55], that signal overlapping was unavoidable. We reasoned that in these conditions the obtained signal overlapping, especially at the longer time points, could not represent actual colocalization. Thus, to prove ICP4 direct binding to the viral folded G4s in cells, we next performed proximity ligation assay (PLA)[56] as the PLA signal is detectable only when two macromolecules of interest are in close proximity (<40 nm), hence directly interacting. The assay was performed in the same conditions as immuno-staining/immuno-DNA-FISH. PLA dots were visible in the nucleus of infected cells (Fig. 2b and Supplementary Fig. S9c). We recorded one spot per nucleus in independent experiments in the majority of PLA positive cells (75%), whereas the remaining (25%) were characterised by two dots per nucleus. This observation perfectly fits with the evidence that just the single parental genome is transcribed in each cell where ICP4 acts as transcription factor[57,58] and that, despite the

high MOI used, only a small portion of cells is infected by multiple parental viral genomes[59]. These data indicate that the essential HSV-1 transcription factor ICP4 directly binds to viral folded G4s in cells during HSV-1 infection.

**Biological role of ICP4-G4 interaction: ICP4 positively reg-ulates its own promoter via G4 interaction.** ICP4 has been reported to function as regulator of HSV-1 gene promoters, including its own[60]. When the promoter regions in between the TATA-box and TSS were analysed, ICP4 was reported to recognise a rather degenerated consensus sequence (i.e. RTCGTCNNYNYSG, where R is purine, Y is pyrimidine, S is C or G and N is any base), while analysis of the full-length promoter regions highlighted several G residues as key elements in ICP4 recruitment and transcription regulation[42,60,61]. We have recently proved that HSV-1 IE promoters, including that of ICP4, are characterised by the presence of multiple and conserved G4s[24]. In particular, from the TATGARAT signal to the ATG start codon, the ICP4 promoter displays three G4s on the leading strand and one on the lagging strand (Fig. 3a). All sequences form parallel-like G4s and display high stability (Supplementary Table S1 and Supplementary Fig. S10a). Given these observations and since our above data indicate a clear ICP4 preference for binding and unfolding parallel G4s, we investigated whether ICP4 recognises its own promoter at the G4 level. We set up a pull-down assay to estimate ICP4 binding towards the G4 sequences embedded in its promoter. The reported ICP4 consensus sequence[42,60,61] was used as positive control; a G-rich non-folding sequence (scram-bled, S) was used as negative control (Supplementary Table S1). ICP4 bound all ICP4-promoter G4s, as well as the reported consensus sequence (Fig. 3a and Supplementary Fig. S10b): the two best bound sequences were ICP4-146532 and ICP4-146666, both of which display high 260/290 values (>50); the other two sequences were also efficiently recognised (low ICP4 signal in the unbound fraction, Fig. 3b) but the quicker dissociation rates, proved by ICP4 signal in the fractions treated with a low strin-gency buffer (ICP4 signal in the wash fraction, Fig. 3a), indicated lower affinity. These two latter G4s have lower 260/290 values (~10). CD analysis confirmed the unfolding activity of ICP4 on its own promoter region: ICP4 reduced molar ellipticity of all the four ICP4 G4 sequences, especially in the region that corresponds to the signal of the parallel G4 topology (i.e. wavelength <270 nm; Fig. 3c).

We next proceeded to assess G4-mediated ICP4 activity on its own expression in cells. To this purpose we cloned the full-length ICP4 promoter sequence (nts 146488–147065 NCBI Reference Sequence: NC_001806.2) upstream of the firefly luciferase gene in a promoterless plasmid. Since the ICP4 promoter contains multiple G4 regions, it was not possible to produce a mutant promoter unable to fold into G4 by single base mutations. We thus generated a delated ICP4 promoter sequence (nts 146629–146778), lacking the 146666 G4 sequence. As already observed with other G4-regulated promoters[28,62], the delated ICP4 sequence showed a remarkably reduced activity upon transfection in U-2 OS cells (Supplementary Fig. S11) and it was thus considered not suitable for further experiments.To evaluated ICP4 activity on its own promoter, we transfected increasing amounts of a mammalian ICP4 expression vector[57] (Fig. 4a and Supplementary Fig. S12) 24 h prior to the ICP4 luciferase reporter vector to allow ICP4 to be expressed. ICP4 promoter activity was remarkably upregulated (2000% at the highest ICP4 concentra-tion) by ICP4 expression, while the TK promoter was only marginally affected, as previously reported[63] (Fig. 4b). The augmented ICP4 promoter activity in the presence of ICP4 is in line with the observed ability of ICP4 to unfold G4s.

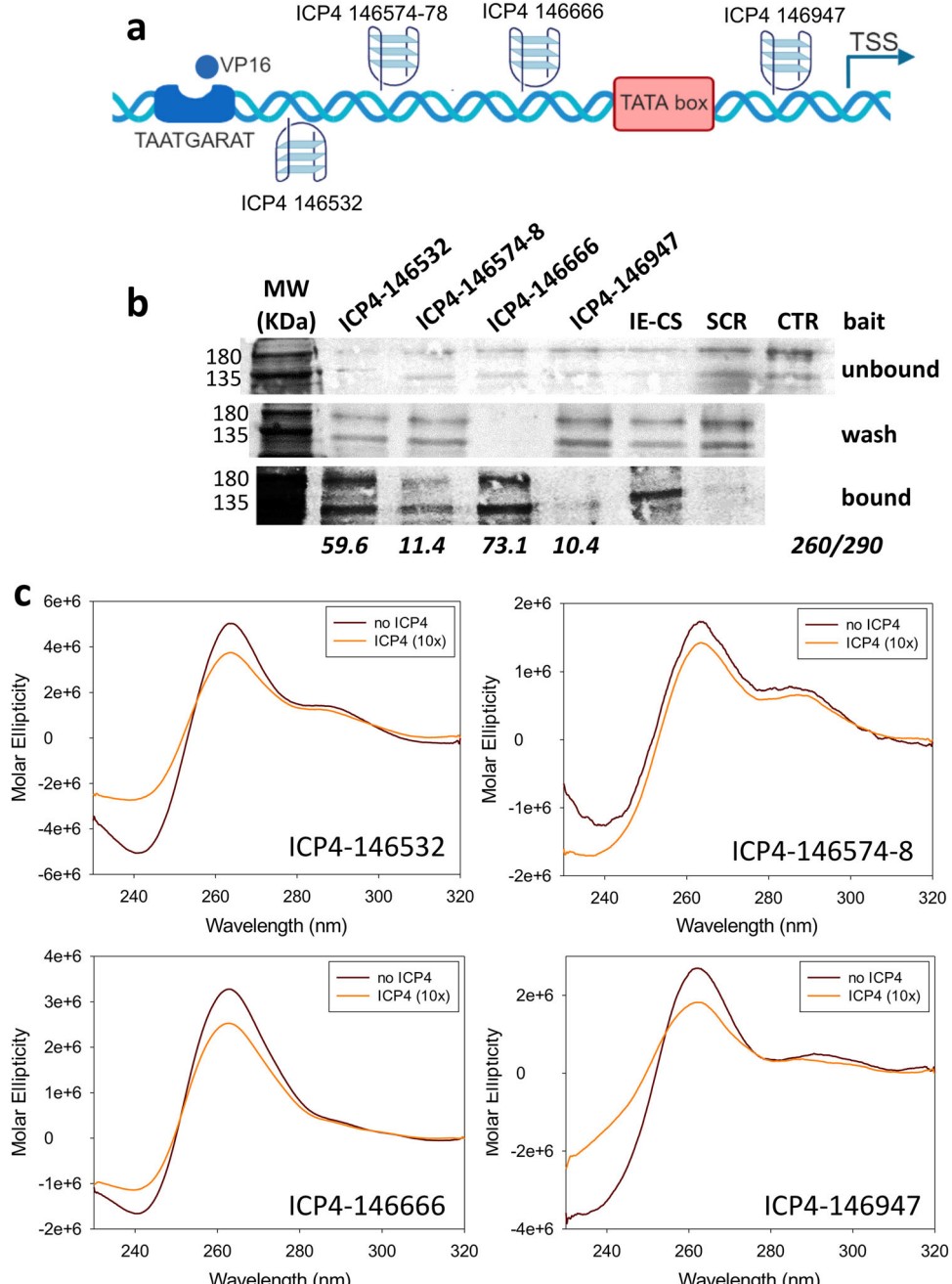

**Fig. 3 ICP4 binding and unfolding of G4 sequences embedded in the ICP4 promoter. a** Schematic representation of G4s present in the ICP4 promoter. Each G4 is indicated with the name of the oligo used in the study. The TAATGARAT region is shown as a blue box with the VP16 protein nearby; the TATA box is represented as a red box; the arrow indicates the position of the Transcriptional Start Site (TSS). The position of each of these features on the double helix with respect to the TSS is on scale. The G4s are shown according to the DNA strand in which they are embedded. **b** Western blot analysis of ICP4 cross-linking pull-down towards the indicated four G4s embedded in the ICP4 promoter. IE-CS is the reported ICP4 consensus sequence[42, 60, 61]; SCR is a G-rich unfolded oligo; CTR is a control lane loaded with infected cell nuclear extracts not subjected to the pull-down procedure. MW: molecular weight (Marker VI, Applichem). The upper panel shows the unbound ICP4 fraction, the middle panel the washed out ICP4 fraction, the lower panel the ICP4 fraction bound to each oligo. The multiband appearance of ICP4 from HSV-1 infected cells in WB has been reported[91] and reflects the presence of differently post-transcriptionally processed isoforms. The experiment was independently performed in duplicate. Each replicate was tested once in each experiment. **c** CD spectra of the indicated G4 sequences embedded in the ICP4 promoter folded in potassium phosphate buffer (20 mM PB, 2 mM KCl) and incubated in the absence/presence of ufICP4 (10x). ICP4 spectrum was subtracted from G4-ICP4 complex spectra. Two independent experiment encompassing one replicate per condition were performed. Representative spectra form one measurement are shown.

To prove the G4-dependence of the ICP4 promoter we analysed its activity in U-2 OS cells in the presence of the G4 ligand B19. A control vector with Renilla luciferase under the HSV-1 TK promoter was used as negative control, as the TK promoter does not contain major putative folding sequences. We observed B19 concentration-dependent inhibition of ICP4 promoter activity, while both TK promoter activity and cell viability were not affected by B19 (Fig. 5a). B19 was able to inhibit

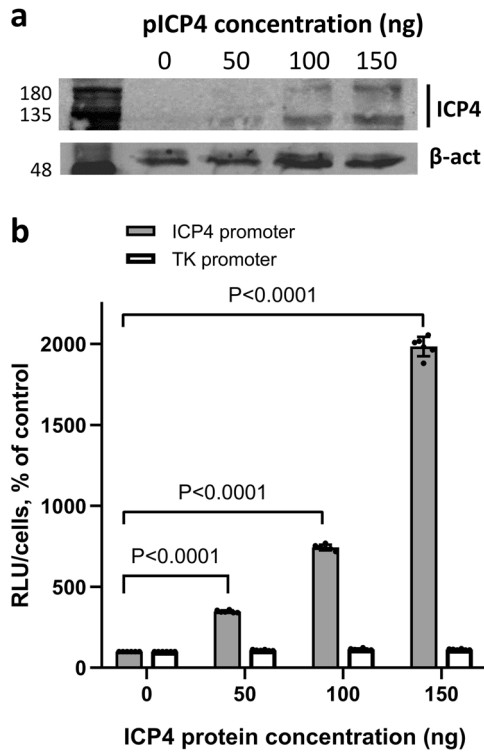

**Fig. 4 G4-mediated ICP4 promoter regulation by ICP4 in cells. a** Western blot analysis of ICP4 expression in the presence of increasing amounts of ICP4 expression plasmid in transfected U-2 OS cells. ICP4 and TK promoters were cloned in the promoterless pGL-4.10 plasmid upstream of the luciferase gene and transfected into U-2 OS cells. β-actin was used as housekeeping control for cell amount. Two independent experiments were performed with one replicate per condition in each experiment. **b** Luciferase reporter assay of ICP4 and TK promoter activity in the presence of increasing amounts of ICP4 protein from ICP4 expression plasmid. Two independent experiments were performed with three replicates per condition in each experiment. *P*-values and SD are reported.

ICP4-mediated increased promoter activity by 10 folds when administered after transfection of the ICP4 promoter plasmid. This inhibitory activity improved up to 65 folds when B19 was administered before transfection of the ICP4 promoter plasmid (Fig. 5b), indicating that B19 effectively competes with ICP4 for the binding to the G4s in the ICP4 promoter. The TK promoter, used as negative control, showed no response to both ICP4 and B19 at both times (Fig. 5b) also in this case. Similar reduction levels were obtained upon treatment with B19 of a plasmid expressing ICP4-YFP (Yellow Fluorescence Protein) under the ICP4 promoter (Fig. 5c and Supplementary Fig. S13). In addition, treatment of HSV-1 infected U-2 OS cells with B19 reduced ICP4 expression at 24 hpi up to 88% of the untreated control (Fig. 5d and Supplementary Fig. S14). The similar inhibition rates obtained in virus- and plasmid-triggered protein expression under the same ICP4 promoter and, presenting different expressed proteins (i.e. native ICP, ICP4-YFP and luciferase), strongly suggest that the G4-ligand B19 inhibits transcription in infected cells acting at the G4s embedded in the ICP4 promoter and point to the presence and functional relevance of G4s in ICP4 promoter in cells.

## Discussion
We have previously shown that viral G4s form massively during HSV-1 replication in cells and that they are present in long repeated region of the HSV-1 genome and in promoters of IE genes[23–25]. The dynamics of cellular and viral G4s in cells have been reported to be regulated by proteins. For instance, folding of one of the best studied G4s, located in the c-myc promoter, is regulated by the interaction with different proteins[64]; similarly, viral G4s that form in the promoter of the HIV-1 integrated genome are processed by cellular proteins that stabilise and unfold them[65,66]. Given these premises, we investigated whether HSV-1 G4s were also processed by proteins during the productive viral life cycle. Besides the two cellular proteins hnRNPA1L2 and NCL already reported to interact with G4s[66–69], we identified for the first time viral proteins able to interact with G4s: these are all involved in some steps of DNA processing. The total number of proteins that selectively interacted with G4s did not increase during infection and, even though they slightly changed in composition, the main proteins recovered at the G4s did not vary along viral replication, supporting the G4-specific role of these proteins. In contrast, composition of the viral proteins that were reported to interact with the whole viral genome reflected the differential protein production during the viral life cycle[22].

ICP4 is a key factor in transcription regulation and viral growth[22,33,60,70]. Up to now, ICP4 had been reported to act as a positive regulator of transcription[70] by recognising an asymmetric DNA consensus sequence, as well as DNA sequences that deviate from the consensus binding site[42]. Our findings indicate that ICP4 exerts its activity also via G4 recognition, as ICP4/viral G4s direct interaction was clearly visible in infected cells throughout the viral cycle.

G4 folding inhibits DNA and RNA polymerases both in vitro and in cell[2]. In fact, when G4s form in gene promoter regions, they typically inhibit gene transcription[28,71]. However, the presence of G4s at gene promoters was shown to be tightly associated with transcription[72]. Our work may help explaining this apparent discrepancy on how G4s are linked to active transcription in cells. ICP4, one of the most important transcription factors in HSV-1 gene expression[33,70], strongly binds to G4s in cells and promotes transcription by unfolding the G4 and promoting assembly of the transcription machinery[70] (Fig. 6). In line with this evidence, we have recently demonstrated that the HSV-1 IE promoters contain multiple conserved G4s, the large majority of which folds into parallel G4s, and ICP4 regulates transcription of at least two of these IE genes (ICP0 and ICP4 itself)[24]. Thus, ICP4 preference for the parallel G4s could be a mechanism for the preferred recognition of promoter-located viral G4s over the large number of G4s that are normally present in the cell[72].

The only natural protein reported to unfold G4s while discriminating among G4 structures is DHX36, a human helicase[73]; three proteins were shown to preferentially bind parallel G4s[9,11,12], while all other G4-interacting proteins up to date have been reported to recognise the folded structure, with no discrimination among different G4 topologies[52]. Recruitment of transcription factors at the G4s may be a mechanism also exploited by the cell: in fact, cellular transcription factors, such as the human mitochondrial transcription factor A and SP1, have been reported to interact with G4s in cells and in vitro, respectively[74–76].

Based on the data collected here, we propose G4s as recognition motifs at the promoter level. When a protein with unfolding properties recognises them and recruits the transcription machinery, such as in the case of ICP4, strong transcription is induced; when a protein stabilizes the G4, induction or repression of transcription can occur, based on the activity of the other proteins that are recruited at the site, if any. Considering that G4-containing promoters corresponded principally to high transcriptional levels in a genome-wide study[72], induction of transcription may be the main role of G4s at gene promoters. In addition, since, except for helicases and few other

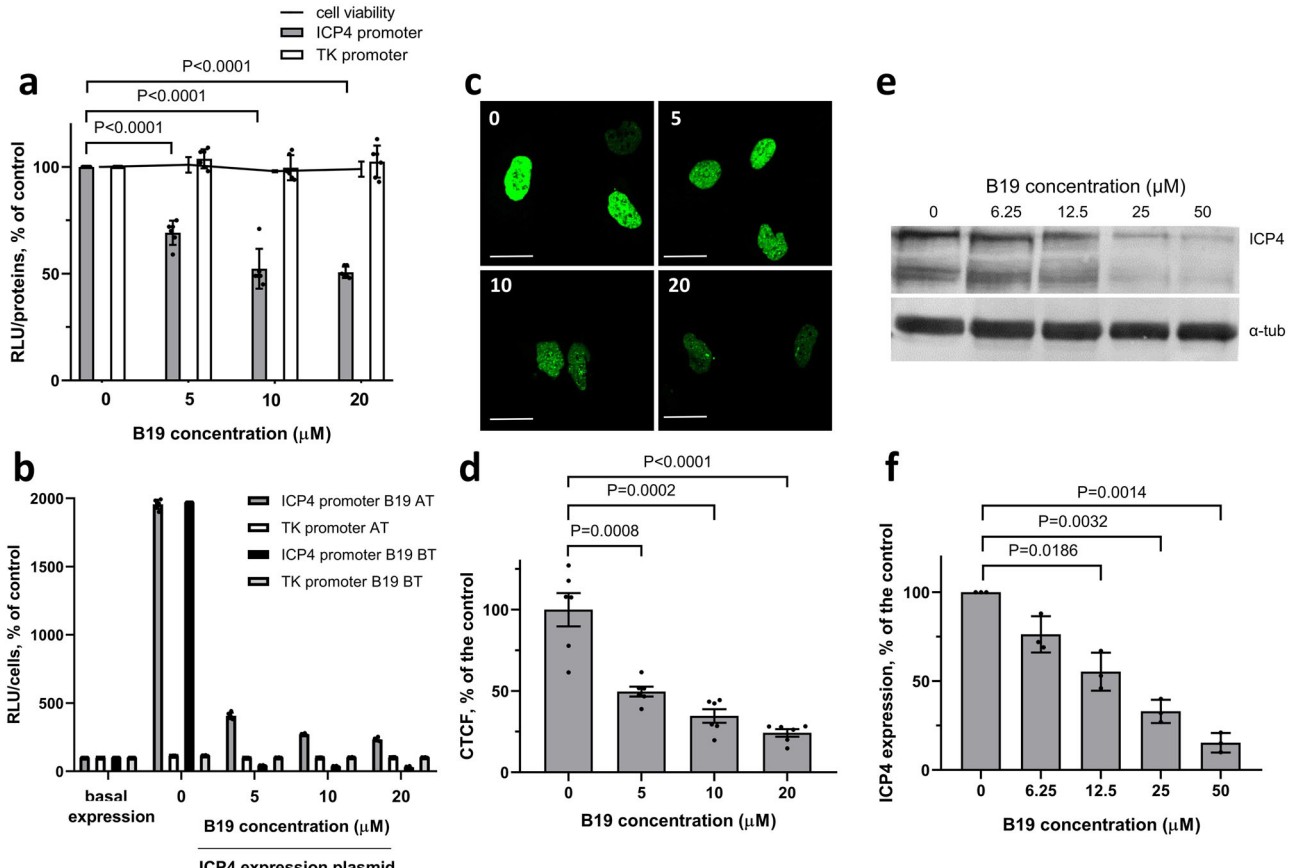

**Fig. 5 ICP4 expression in the presence of the G4-ligand B19. a** Luciferase reporter assay of ICP4 promoter and control TK promoter activity in the presence of increasing B19 concentrations. ICP4 and TK promoters were cloned in the promoterless pGL-4.10 plasmid upstream of the luciferase gene and transfected into U-2 OS cells. U-2 OS cell viability in the presence of increasing concentrations of B19 was also evaluated. Two independent experiments with three replicates per condition were performed. Significant *P*-values and SD are reported. **b** Luciferase reporter assay of ICP4 and TK promoter activity in the presence of ICP4 expression plasmid and in the presence/absence of B19 added 2 h after (AT) or before (BT) transfection of reporter plasmid. Mean average values of two independent experiments with three replicates per condition are reported. SD is indicated. **c** Representative confocal microscopy images of U-2 OS cells transfected with ICP4-EYFP plasmid that express EYFP under the ICP4 promoter, in the presence of increasing concentrations of B19 (0, 5, 10, 20 μM, as indicated). Scale bars 20 μm. **d** Quantification expressed as corrected total cell fluorescence (CTCF) of cell images, calculated on 6–8 slides (50–100 nuclei), mean average levels from two independent experiments. P-values and SD are reported. **e** Western blot analysis and **f** quantification of the viral ICP4 protein in the presence of increasing concentrations of B19 in U-2 OS cells infected with HSV-1 strain F (MOI 1). MW molecular weight (Marker VI, Applichem). ICP4 bands were quantified with respect to alpha-tubulin (housekeeping) levels in each lane. Mean average values of three independent experiments with one replicate per condition are reported. Significant *P*-values and SD are indicated.

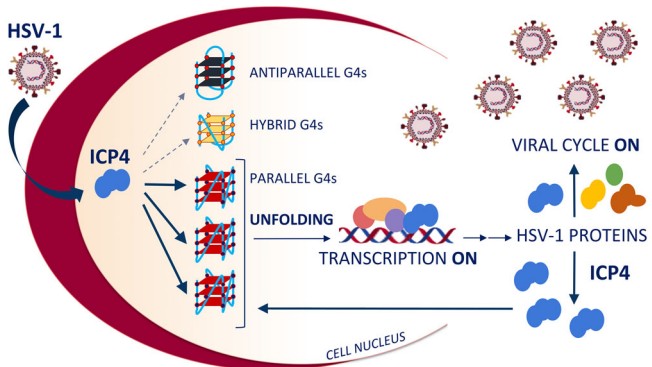

**Fig. 6 Model of ICP4 activity at G4s in viral IE promoters in HSV-1 infected cells.** Schematic of the herpesvirus transcription factor ICP4 regulating viral gene expression and boosting its own expression through binding to parallel DNA G-quadruplexes (G4s).

proteins[21,73,77–79], the vast majority of G4 binding protein stabilizes G4s[80], recruitment of transcription factors by G4-interacting proteins may be the most common mechanism.

Our data indicate that ICP4 can also recognise cellular parallel G4s, therefore suggesting a role of HSV-1 in human gene expression regulation. In fact, ICP4 was reported to upregulate the vascular endothelial growth factor (VEGF), through binding of ICP4 to the three tracts of GC-rich sequences in the proximal human VEGF promoter[81]. The authors did not describe the specific ICP4-VEGF interaction but hinted that formation of G4s at that specific region of the VEGF promoter could be targeted by G4 ligands to suppress transcription, to be exploited as anticancer strategy[82]. Notably, the G4 embedded in the VEGF promoter displays a parallel structure. These data altogether indicate that the G4-mediated mechanism of transcription regulation is conserved and exploited by both eukaryotic cells and viruses and suggest that cross regulation may occur via this mechanism.

We have previously indicated the possible use of G4 ligands as antiviral compounds. Indeed, B19 and other G4 ligands were

proven to hinder HSV-1 viral replication without affecting HSV-1 entry[23,26]. Here we further revealed that B19 strongly affects the expression of ICP4 protein acting on its own promoter, thus hampering all subsequent ICP4 transcriptional activities. Thus, the described data may offer a new direction for antiviral design. Up to date, only polymerase inhibitors are available against HSV-1[83]. The rising of resistant HSV-1 strains that may be extremely dangerous especially in transplant and immunocompromised patients urges the discovery of new antivirals with a different mechanism of action. We propose that selective anti-ICP4 promoter G4-ligands would hamper the viral cycle at a very early stage, preventing neuronal damage linked to HSV-1 infection.

## Methods

**Cell lines and viruses.** Human bone osteosarcoma cells (U-2 OS, European Collection of Authenticated Cell Cultures (ECACC), purchased from Sigma Aldrich, Milan, Italy) were maintained in Dulbecco's Modified Eagle Medium (DMEM, Gibco, Life Technologies, Monza, Italy) supplemented with 10% heat-inactivated and filtrated foetal bovine serum (FBS, Gibco, Life Technologies, Monza, Italy). Cells were maintained in a humidified incubator set at 37 °C with 5% $CO_2$ and monthly checked for mycoplasma contamination. HSV-1 strain F (GenBank: GU734771) was propagated on U-2 OS cells. Viral stocks were prepared in 3% FBS Dulbecco's modified Eagle's media (Gibco, ThermoFisher Scientific, Monza, Italy).

**Circular dichroism spectroscopy.** DNA oligonucleotides were diluted to a final concentration (4 µM) in potassium phosphate buffer (PB, 20 mM, pH 7.4, KCl 2–80 mM). All samples were annealed at 95 °C for 5 min and gradually cooled to room temperature. CD spectra were recorded on a Chirascan-Plus (Applied Photophysics, Leatherhead, UK) equipped with a Peltier temperature controller using a quartz cell of 5 mm optical path length, over a wavelength range of 230–320 nm. For the determination of $T_m$, spectra were recorded over a temperature range of 20–90 °C, with temperature increase of 5 °C. The reported spectra are baseline-corrected for signal contributions due to the buffer. Observed ellipticities were converted to mean residue ellipticity $(\theta) = \deg \times cm^2 \times dmol^{-1}$ (mol ellip). $T_m$ values were calculated according to the van't Hoff equation, applied for a two-state transition from a folded to unfolded state, assuming that the heat capacity of the folded and unfolded states is equal. All oligonucleotides were tested at least twice in independent experiments. All measures were repeated at least twice.

**Nuclear proteins extraction from HSV-1 infected cells.** U-2 OS cells ($5.3 \times 10^6$) were plated in 150 cm$^2$ cell culture flasks in DMEM supplemented with 10% FBS. In all, 24 h after seeding, the medium was removed and cells were infected with HSV-1 strain F (MOI of 2) in serum-free medium. After 1 h at 37 °C, cells were washed with 1X PBS and fresh complete medium was restored. At different times post-infection (8 and 16 h.p.i.), cells were scraped off and centrifuged at 1250 rpm for 5 min at 4 °C. Pellets were processed according to the B protocol of the NXTRACT kit (CelLytic™ NuCLEAR™ Sigma-Aldrich, Milan, Italy) https://www.sigmaaldrich.com/content/dam/sigma-aldrich/docs/Sigma/Bulletin/nxtractbul.pdf), in order to obtain native nuclear proteins. Proteins extracts were quantified using the Pierce™ BCA Protein Assay kit (ThermoFisher Scientific, Monza, Italy) and stored in aliquots at −80 °C until use.

**Pull-down assay.** The pull-down assay was performed in order to identify HSV-1 G4-protein interactions. Dynabeads→ M-280 Streptavidin (Life Technologies, Monza, Italy) were used to immobilise the biotinylated oligonucleotides, according to the manufacturer's protocol. Briefly, 50 µL of beads were incubated with 300 pmol of biotinylated oligonucleotides (15 min at room temperature, RT). Control samples without oligonucleotides were included in each experiment in order to monitor the nonspecific binding to the beads. After three washes (50 mM Tris-HCl, 150 mM NaCl), nuclear proteins extracts (5 µg) were incubated with the immobilised DNA fragments and maintained at 4 °C for 2 h. To wash out nonspecific binders, samples were rinsed three times in 20 mM phosphate buffer (PB) and 150 mM KCl. Two proteins elution were performed: the first in NaCl 2 M (10 min) and the second in 2x Laemmli buffer (4% SDS, 80% glycerol, 120 mM Tris-HCl pH 6.8, 200 mM DTT, 0.02% bromophenol blue, 10 min). Eluted proteins were denatured at 95 °C and separated on SDS–PAGE. Gels were stained overnight in colloidal Coomassie staining (0.02% w/v CBB G-250, 5% w/v aluminium sulfate-(14–18)-hydrate, 10% v/v ethanol, 2% g/v orthophosphoric acid) and unstained in 10% ethanol and 2% orthophosphoric acid for at least 30 min. Destained gels (10% acetic acid in milliQ water) were cut into bands of ~0.5 cm per side. Each band was further fragmented and subjected to tryptic digestion.

**Bands digestion and LC-MS protein identification.** Bands were treated according to established in-gel digestion protocols[84]. Briefly, they were first bleached with 50% $CH_3OH$ and 2.5% acetic acid and dehydrated with $CH_3CN$. Peptides were then reduced with 50 µL of DTT (10 mM in 100 mM NH$_4$HCO$_3$) for 30 min at RT. After DTT removal, peptides were treated with 50 µL of iodoacetamide (50 mM in 100 mM NH$_4$HCO$_3$) for 30 min at RT in the dark in order to alkylate cysteine residues. Bands were washed with 100 mM NH$_4$HCO$_3$ and dehydrated with $CH_3CN$ twice. After that, 1 µg aliquot of MS-grade trypsin (ThermoFisher Scientific, Monza, Italy) in 50 µL of 50 mM NH$_4$HCO$_3$ was added to each dehydrated bands and incubated on ice for 30 min. The excess of trypsin was eliminated and substituted with 40 µL of 50 mM NH$_4$HCO$_3$. Samples were then incubated overnight at 37 °C. Peptides were extracted twice with 5% HCOOH and two more times with 50% $CH_3CN$, 5% HCOOH. Samples were further concentrated in SpeedVac (Hetovac VR-1, Heto Lab Equipment, Denmark) to 1 µL and resuspended in 20 µL of 1% HCOOH. Peptides were then analysed by liquid chromatography mass spectrometry (LC-MS) using an Agilent 1290 Infinity HPLC (Agilent Technologies, Santa Clara, CA, USA) connected to a Xevo G2-XS QTof mass spectrometer (Waters, Milford, MS, USA) from Micromass (Manchester, UK). A full scan mode and a high energy MS$^e$ scan mode were used. Parent ions having the charge state 4$^+$, 3$^+$ and 2$^+$ and signals more intense than 200 counts and the related MS$^e$ fragments ion signals more intense than 50 counts were subjected to the[85] Database Search to identify their parent protein.

**ICP4 purification by immunoprecipitation.** ICP4 was purified by means of the Cross-link IP Kit (ThermoFisher Scientific, Monza, Italy), according to manufacturer instructions. Briefly, 10 µg of anti-ICP4 antibody (mouse monoclonal H943, Santa Cruz Biotechnology, Dallas, TX, USA) was coupled to Protein A/G Plus Agarose for 1 h and subsequently cross-linked for 30 min (following procedures described in section A and B of the protocol, Pierce® Cross-link Immunoprecipitation Kit Manual). ICP4 immunoprecipitation (section E) was conducted incubating 10 µg of infected U-2 OS cells nuclear extract (see Nuclear proteins extraction from HSV-1 infected cells) overnight at 4 °C under gentle shaking. ICP4 was eluted with the provided elution buffer (as indicated in section F). The acid pH of the elution buffer was neutralised adding 5 µl of 1 M Tris-HCl pH 9.5 to the sample, as specified in the purification protocol. The eluted fraction was quantified using the Pierce™ BCA Protein Assay kit (ThermoFisher Scientific, Monza, Italy) whereas protein integrity was checked by WB.

**ICP4 purification by ultracentrifugation.** Purification of ICP4 by ultracentrifugation was performed as previously described[44]. Briefly, $5.3 \times 10^6$ plated U-2 OS cells were infected with HSV-1 strain F at an MOI of 2 in serum-free medium. After 1 h at 37 °C, cells were washed with 1X PBS and fresh complete medium was restored. Cells were scraped off 6 hpi and centrifuged at 1250 rpm for 5 min at 4 °C. Pellets were processed according to the NXTRACT kit (CelLytic™ NuCLEAR™ Sigma-Aldrich, Milan, Italy, protocol B), to obtain native and functional nuclear proteins: cell pellets were incubated with 1X hypotonic lysis buffer (10 mM HEPES, pH 7.9, 1.5 mM MgCl$_2$, 10 mM KCl, 1 mM DTT, 1% protease inhibitor cocktail) for 5 min at 4 °C, allowing cells to swell. The suspended cells were centrifuged for 5 min (420 × g at 4 °C). Supernatant was discarded and the obtained pellet was resuspended in 1X hypotonic lysis buffer. The cells were mechanically disrupted and centrifuged for 20 min at 10,000 × g at 4 °C. The supernatant (cytoplasmic fraction) was transferred to a new tube while the crude nuclei pellet was resuspended in 80–120 µL of extraction buffer (20 mM HEPES, pH 7.9, 1.5 mM MgCl$_2$, 420 mM NaCl, 0.2 mM EDTA, 25% (v/v) glycerol, 1 mM DTT, 1% protease inhibitor cocktail), to allow nuclear proteins isolation. After 30 min of incubation at 4 °C, the sample was centrifuged for 5 min (17,000 × g at 4 °C) and the supernatant was preserved into a clean chilled tube. The purification of ICP4 was carried out applying the nuclear proteins extract into the Amicon Ultra – 0.5 mL centrifugal filter unit equipped with a membrane of 100 K nominal molecular weight limits (NMWL), and subsequently washing with potassium phosphate buffer (20 mM PB and 100 mM KCl) According to the manufacturer's protocol, the sample was centrifuged at 14,000 × g for 10 min (at 4 °C). The purified sample was recovered spinning for 2 min at 1000 × g and finally quantified using the Pierce™ BCA Protein Assay kit (ThermoFisher Scientific, Monza, Italy) and stored in aliquots at −80 °C until use. Molecular weight purified fractions were validated through LC-MS analysis, ICP4 was the highest covered protein, with negligible traces of cellular factors, like Filamin-A retaining minimal scores.

**G4-binding proteins cross-linking assay.** Protein nuclear extracts were obtained as described in the pull-down experiment. Biotinylated oligonucleotides 150 pmol, folded in 20 mM PB pH 7.4, 80 mM, KCl, were bound to 50 µl of streptavidin-coated magnetic beads. DNA coupled-beads were activated with formaldehyde (Sigma-Aldrich, Milan, Italy)[86] 5% for 15 min at RT and then incubated with nuclear proteins 15 µg extract at 4 °C for 45 min and proteins excess was washed withTris-HCl pH 7.5 50 mM, 150 mM NaCl solution. Samples were then analysed by western blot analysis, with an anti-ICP4 antibody (mouse monoclonal H943 Santa Cruz). Briefly, samples were electrophoresed on a 8% SDS–PAGE and transferred to a nitrocellulose blotting membrane (Amersham TM Protan TM, GE Healtcare Life science, Milan, Italy) by using trans-blot SD semi-dry transfer cell (Bio-Rad Laboratories, Milan, Italy). The membrane was blocked with 2.5% skim milk in phosphate-buffered saline solution, incubated with the anti-ICP4 antibody

(1:100) and then with the ECL Plex Goat-α-Mouse IgG-Cy5 (GE Healthcar-eLifesciences, Milan, Italy). Images were captured on Typhoon FLA 9000.

**FRET and FRET-melting assays.** For Föster Resonance Energy Transfer (FRET) experiments, oligonucleotides were diluted to 0.25 μM in 20 mM PB pH 7.4 and 2–80 mM KCl, heat denatured for 5 min at 95 °C, and allowed to cool to RT. For the creation of the double strand sequence (ds) the G-rich and the C-rich oligos were added to the same sample (1:1 ratio) and heat denatured for 5 min at 95 °C. For samples that were tested with the complementary strand, the C-rich oligo was added to the sample 18 h after denaturation of the G-rich oligo. Samples were incubated alone or with ipICP4/pICP4 or BSA for 1 h at 4 °C and subsequently fluorescence intensity was measured in a LightCycler II (Roche, Milan, Italy) by observing 6-carboxyfluorescein (6-FAM) emission. Protein buffers (50 mM Tris-HCl, 150 mM NaCl or 20 mM PB, 100 mM KCl) were added to the oligos, depending on the experiment. The excitation wavelength was set at 480 nm and the emission was recorded from 500 to 650 nm. In the melting assay, fluorescence was monitored from 30 °C to 95 °C (1 °C/min).

**Un2 and G4 visualisation in infected cells.** U-2 OS cells were plated ($2.75 \times 10^5$) on glass coverslips and infected, with HSV-1. Cells were fixed, 8 hpi, with 2% paraformaldehyde 2% sucrose in PBS 1X for 10 min and permeabilized with 0.1 % Tween-20, for 45 min at RT. Cells were treated with RNAse A (40 μg/ml) for 45 min at RT. A quantity of 50 ng of UN2 shifted pla-btn probe (Supplementary Table S1) was denatured in hybridisation buffer (10% formamide; 1X Saline Sodium Citrate (SSC) buffer) for 5 min at 95 °C and immediately cooled down on ice. The hybridisation reaction on coverslips was carried out with an initial step of 5 min at 85 °C and overnight incubation at 37 °C. Coverslips were treated with BlockAid™ (ThermoFisher Scientific, Monza, Italy) for 1 h at RT, prior to incubation with the anti-G4 antibody (1 μg/μl; 1H6, kindly gifted from Prof. Lansdorp PM) and the anti-biotin antibody (1:1000 dilution, Bethyl Laboratories, Inc. Montgomery, TX, USA) for 1 h at RT. To note that, even if anti-G4 antibodies, such as 1H6 and BG4, are unvaluable tools to study G4s in cells, they all show some degree of unspecificity[87,88]. Secondary antibodies anti-mouse Alexa 488 and anti-rabbit Alexa 546 (1:300 dilution, ThermoFisher Scientific, Monza, Italy) were incubated for 1 h at RT. All washes were carried out in 1x PBS - 0.05% Tween 20.

**Proximity ligation assay.** U-2 OS cells were plated ($2.75 \times 10^5$) on glass coverslips and infected with HSV-1. Infected cells were fixed, at 8 hpi, with 2% paraformaldehyde 2% sucrose in PBS 1X for 10 min and permeabilized with 0.4% Triton X-100, for 5 min at RT, and treated with RNAse A (40 μg/ml) for 45 min at RT, 24 h after seeding A quantity of 50 ng of UN2 shifted pla-btn probe (Supplementary Table S1) was denatured in hybridisation buffer (10% formamide; 1X Saline Sodium Citrate (SSC) buffer) for 5 min at 95 °C and immediately cooled down on ice. The hybridisation reaction on coverslips was carried out with an initial step of 5 min at 85 °C and overnight incubation at 37 °C. Coverslips were treated with BlockAid™ (ThermoFisher Scientific, Monza, Italy) for 1 h at RT, followed by 2 h incubation at RT with the primary antibodies (anti-biotin 1:1000 Bethyl - and anti-ICP4 H943 1:250 Santa Cruz Biotechnology, Inc., Dallas, TX, USA). The PLA assay was carried out with the Duolink PLA in situ kit, PLA probe anti-rabbit plus, the Duolink in situ PLA probe anti-mouse MINUS and the in situ detection reagent Green (Sigma-Aldrich, Milan, Italy) following the manufacturer's protocol. Nuclei were stained with TOTO3 (1:2000, ThermoFisher Scientific, Monza, Italy) for 20 min at RT. PLA results were visualised using a Leica TCS SP2 confocal microscope (×60 objective). All the PLA experiments were performed at least three times independently. For each PLA experiment, the following controls were used: mock cells, infected cells w/o probe, infected cells w/o antibodies.

**Cellular cytotoxicity.** Cytotoxic effects were determined by MTT assay. U-2 OS cells were grown and maintained according to manufacturer's instructions (https://www.lgcstandards-atcc.org). Cells were plated into 96-microwell plates to a final volume of 100 μL and allowed an overnight period for attachment. The following day, the tested compound (B19) was added to each well and tested in triplicate. Control cells were treated in the exact same conditions. Cell survival was evaluated by MTT assay, 24 h after treatment: 10 μL of freshly dissolved solution of MTT (5 mg/mL in PBS) were added to each well, and after 4 h of incubation, MTT crystals were solubilised in solubilisation solution (10% sodium dodecyl sulphate (SDS) and 0.01 M HCl). After overnight incubation at 37 °C, absorbance was read at 540 nm. Data were expressed as mean values of at least three individual experiments conducted in triplicate. The percentage of cell survival was calculated as follows: cell survival = $(A_{well} - A_{blank})/(A_{control} - A_{blank}) \times 100$, where blank denotes the medium without cells. Each experiment was repeated at least three times

**Reporter assays.** Vectors pGL4.10-ICP4 and pGL4.74 TK (150 ng each) were transfected in $1.2 \times 10^5$ U-2 OS cells per well onto 12-well plates, using Lipo3000 transfection reagent (Invitrogen, Life Technologies Italia, Monza, Italy). B19 was added to the cell medium 2 h prior/after transfection at increasing concentrations (5–20 μM), to avoid interference, if any, with transfection. Expression of firefly luciferase was determined 24 h after transfection using the Britelite plus Reporter

Gene Assay System (PerkinElmer Inc., Milan, Italy) at a Victor X2 multilabel plate reader (PerkinElmer Inc., Milan, Italy), according to the manufacturer's instructions. Cells were lysed in 0.1% Triton-X100-PBS and protein concentration was determined by BCA assay (Thermo Scientific Pierce, Monza, Italy). Luciferase signals were subsequently normalised to total protein content or cell content, according to the manufacturer's protocol (http://ita.promega.com/~/pdf/resources/pubhub/cellnotes/normalising-genetic-reporter-assays/). Each assay was performed in duplicate and each set of experiments was repeated at least three times.

**ICP4-YFP visualisation in cells.** U-2 OS cells were plated on glass coverslips ($2.75 \times 10^5$). Cells were transfected with pICP4-YFP[89] (750 ng/well; kindly gifted by Prof. Roger Everett), 24 h post seeding. Cells were fixed with 2% paraformaldehyde 2% sucrose in PBS 1X, 24 h post transfection and visualised at ×40 and ×60 objective with a Nikon A1Rsi+ Laser Scanning confocal microscope equipped with NIS-Elements Advanced Research software (Nikon Instruments Inc., Melville, USA).

For quantification of ICP4-YFP protein expression, the 488-nm laser (488 nm excitation and 500–550 nm emission filters) was used with the laser power that was set on the untreated control sample and kept unchanged for the acquisition of all pictures. The fluorescent images were then analysed using the ImageJ Software by selecting one cell at a time and measuring the area, integrated density and mean grey value. Using the calculation for corrected total cell fluorescence (CTCF) = integrated density–(area of selected cell × mean fluorescence of background readings), the fluorescence intensity of each cell was calculated. For each image, a background area was used to normalise against auto-fluorescence. For each experimental condition 6–8 images were acquired with a ×40 objective, and all cells per slide were analysed.

**B19 treatment of HSV-1 infected cells.** U-2 OS cells were plated onto six-wells plates ($2.75 \times 10^5$). Cells were infected with HSV-1 (MOI of 1) 24 h post seeding and treated with increasing concentration of B19 (3.125–50 μM), 1 hpi. Cells were collected 24 hpi and cellular/viral proteins were extracted in RIPA buffer (150 mM NaCl, 5 mM EDTA pH 8, 50 mM Tris-HCl pH 8, 0.5% Igepal CA-630, 1x Protease Phosphatase Inhibitor CockTail). Protein concentration was determined and samples (15 μg) were denatured at 95 °C prior to immunoblot analysis.

**Immunoblot analysis.** Immunoblot analysis were performed on 8–10% SDS–PAGE and transferred to a nitrocellulose blotting membrane (Amersham TM Protan TM, GE Healtcare Life science, Milan, Italy) by using trans-blot SD semi-dry transfer cell (Bio-Rad Laboratories, Milan, Italy). The membranes were blocked with 2.5% skim milk in PBST (0.05% Tween 20 in PBS). Membranes were incubated with the respective primary antibody directed against ICP4 (mouse monoclonal H943; Santa Cruz Biotechnology, Dallas, TX, USA), alpha-tubulin (mouse monoclonal; Sigma-Aldrich, Milan, Italy). After three washes in PBST, membranes were incubated with ECL Plex Goat-α-Mouse IgG-Cy5 (GE Healthcare Life sciences, Milan, Italy). Images were captured on the Typhoon FLA 9000. For quantification purposes the ImageJ software was used. For each sample ICP4 expression values were normalised on the relative housekeeping signal (alpha-tubulin).

**ICP8 purification from BL21 bacterial cells.** A colony BL21 (DE3) pLysS expressing ICP8 was inoculated in LB broth supplemented with Amp (50 μg/ml) and Cam (34 μg/ml) and incubated at 30 °C overnight shaking. The culture was diluted 1:100 in antibiotic-rich LB medium and incubated at 37 °C shaking until the OD600 was 0.7–0.8. UL29 gene expression was induced by addition of 0.75 mM isopropyl-β-D-thiogalattoside (IPTG) and 4 μM ZnOAc to the medium. Bacteria were incubated at 16 °C overnight shaking. The samples were centrifuged at 2500 × g for 10 min at 4 °C, then the pellet was resuspended in 100 μl of lysis buffer (1X phosphate-buffered saline (PBS), 1 mM PMSF, 50 μM ZnOAc, 1 mg/ml lysozyme). The suspended cells were disrupted via freezing-and-thawing cycles, alterning 5 min in liquid nitrogen with 5 min in thermostatic bath at 42 °C, repeated four times. The samples were vortexed and then centrifuged at 13,000 rpm for 2 min. ICP8 protein was purified via GST-precipitation, using Pierce Glutathione Agarose beads (Thermo Fisher Scientific Inc. Milan, Italy). The equilibrated resin was incubated with the bacterial lysate at 4 °C overnight. The washing phase was followed by the elution of the protein from the beads, by adding 1 resin-bed volume of Elution Buffer (composed of 50 mM Tris HCl pH 8, 150 mM NaCl, 10 mM Glutathione) on the resin. After an incubation of 10 min on ice, the tube was centrifuged at 0.7 rcf for 2 min. Protein aliquotes were stored in 20% (w/v) glycerol at −80 °C.

**ICP8 electrophoresis mobility shift assay (EMSA).** The λ32P-labelled oligonucleotides were induced to fold into G-quadruplex structure via incubation in a mix containing 10 mM Lithium cacodylate pH 7.2 and 150 mM NaCl or 50 mM KCl up to a final volume of 80 μl. The mix was heated at 95 °C for 5 min and cooled down at room temperature. The binding reaction was performed at 4 °C for 30 min in 20 mM Tris–HCl, pH 8, 30 mM KCl, 1.5 mM MgCl₂, 1 mM ZnOAc, 8% glycerol, 1% Phosphatase Inhibitor Coktail I, 5 nM NaF, 1 mM Na3VO4, 2.5 ng/ml poly dI-dC. The analyses were carried out in 8% polyacrylamide gel (29:1) in TBE supplemented with 150 mM NaCl or 150 mM KCl.

**ICP4 electrophoresis mobility shift assay**. Fluorescent labelled oligos were folded in 20 mM PB, pH 7.4 and KCl. The binding reaction was carried out at 4 °C for 45 min in 20 mM Tris–HCl, pH 8, 25 mM KCl, 1.5 mM MgCl$_2$, 8% glycerol, 1% Phosphatase Inhibitor Coktaill. The analyses were carried out in 5% polyacrylamide gel (29:1) in 50 mM PB, pH 7.4.

**Statistics and reproducibility**. For each experiment, the sample size $n$ refers to the number of independent experiments; the number of replicates per experiment is also stated in the figure legends. The replicates indicate the number of samples of each condition examined in each experiment. Independent experiments indicate the number of assays performed, for each described test, from the sample preparation to the final described result, i.e., for the immunofluorescence from the cell harvesting up to the final image detection at the confocal microscope. All $P$ values were calculated using an unpaired two-tailed Student's $t$ test and the significance for each sample has been indicated using the calculated $P$ value. $P$ values were not calculated for datasets with $n < 3$. All the statistical analysis were performed using GraphPad Prism 8. The error bars indicate the s.d., as stated in the figure legends.

**Reporting summary**. Further information on research design is available in the Nature Research Reporting Summary linked to this article.

## Data availability

All data generated or analysed during this study are included in this published article (and its supplementary information files). The mass spectrometry proteomics data have been deposited to the ProteomeXchange Consortium via the PRIDE partner repository[90] with the data set identifier PXD024743. Uncropped original gel images of Figs. 3b, 4a and 5d in the main text are shown in Supplementary Fig. S14. Source data for the main and supplementary figures is provided in Supplementary Data 1.

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

## Acknowledgements

This work was supported by the University of Padua (PRID/SID 2018). We thank Dr. Emanuela Ruggiero for help in designing the TOC, Beatrice Tosoni for assistance in WB assays, Elena Trallori for ICP8 purification from bacterial cells and Prof. Arianna Calistri for helpful discussion on HSV-1.

## Author contributions

Conceptualization by I.F. and S.N.R.; methodology by I.F. and M.N.; investigation by I.F., P.S., M.N. and S.L.; writing – original draft by I.F.; writing – review & editing by S.N.R.; funding acquisition by S.N.R.

## Competing interests

The authors declare no competing interests.
