## [Peer Review File · Communications Biology]

Reviewers' comments:

Reviewer #1 (Remarks to the Author):

Frasson et al reported ICP4, the major HSV-1 transcription factor binds to and unfolds several viral G4s, and ICP4, through G4 binding, regulates its own promoter.

However, the work remains preliminary as definitive demonstration of a preference for a specific G4 topology is not demonstrated. In addition, the data supporting the unfolding of G4s is not convincing as currently presented.

1. The analysis performed in Figure 1 does not include a G4 of antiparallel topology, only parallel and mixed topology. Inclusion of a G4 of anti-parallel topology would strengthen their claim in line 179. Why was hTel-T (antiparallel from the CD presented in Fig S3A not used in this binding assay?). Was an i-motif tested – un2 complementary? Where EMSAs also performed?

2. The graph in Figure 1B does not add to the immunoblot presented. The control used is an input control, a percentage of the nuclear extract added before each pulldown. An oligonucleotide that does not bind ICP4 would be more appropriate as a negative control, and then relative binding could be graphed if desired.

3. ICP4 used in the FRET unfolding assays was from IP and ultrafiltration, and its purity determined by mass spectrometry and ICP4 immunoblot. Were these samples separated by SDS PAGE and stained for total protein? The source of the protein influenced the results presented in Table S3 (for un2 it was at least 3-fold). While mentioned, using bacterially expressed and purified ICP4 would remove these problems and the influence of other proteins.

4. The presentation of the FRET unfolding assays in Figure 1C and S3B was confusing and a diagram or clearer explanation of the experimental procedure should be provided. Line 156/157 'In the case of un2 and myc, addition of ipICP4 induced unfolding, the degree of which largely or mildly increased when the G4 complementary strand was added before or concurrently to ipICP4, respectively (Table S3 and Figure 1C)'. Does Figure 1C have the complementary strand present? Figure S3B shows the unfolding assay when complementary strand was added at the same time as ICP4.

Why was the FRET unfolding assay not done at 37C? Does it require ATP?

These findings would be strengthened by demonstrating the unwinding by another method such as the in vitro unwinding experiments performed with native gels as shown in Fig 2F and H of Ribeyre, et al, The Yeast Pif1 Helicase Prevents Genomic Instability Caused by G-Quadruplex-Forming CEB1 Sequences In Vivo. Plos Genetics 2009, 5 (5), e1000475. doi:10.1371/journal.pgen.1000475.

5. In addition, the FRET unfolding assays in Figure 1C do not to reflect the results from the CD (Figure 1D). Is the small decrease in the 260nm peak observed in the CD presented in Fig 1D and 3B, just dilution from the addition of protein?

6. The use of the proximity ligation assay to visualise a signal G4 bound to ICP4 will be of interest to others in the field. Was an immuno-FISH performed with un2 and anti-ICP4 also?

7 What is the sequence of the IE-CS used in Figure 3A (as it is not listed in Table S1) in the pull down. ICP4 binds as strongly to IE-CS as some of the parallel promoter G4s. Is it a double stranded DNA consensus sequence for ICP4? The ability of ICP4 to bind the G4 does not reflect the unwinding observed in Figure 3. Also, these are all parallel G4s, with 146574-8 (Fig S7) showing some mixed topology.

8. A schematic diagram of the ICP4 promoter showing the location of each of the four G4s would assist with the interpretation of the reporter assays in Figure 4 and 5. The statement in line 216, indicates some of the G4s are downstream of the transcription start site?

Inclusion of a ICP4 promoter reporter construct with a mutant G4 binding sequence that could not form a G4 (ie of 14666), would strengthen the claim that ICP4 G4 unfolding is required for ICP4

activation of the promoter.

9. Figure 5C shows very consistent expression of the ICP4-EYFP plasmid across the concentrations of B19 used. Given the plasmid was transiently transfected into the U-2 OS cells, how was the usual variation in expression between transfected cells overcome in collecting this data? A wider field of cells or lower magnification should be shown to be more representative.

10. The immunoblot presented in Figure 5C is not convincing given the uneven bands of tubulin. Also, was the same amount of total protein loaded as the intensity of the variation in the bands has a major influence on the interpretation of the level of ICP4 expression in response to B19 concentration.

11. The statement on line 333 is incorrect, one example of a protein binding to G4s in a topology specific manner is described in reference 11.

12. Given factors such as ICP4 are proposed to bind and unfold G4s within promoter regions to bring in other factors and the PIC. What is the role of the unwinding? From their location is it site exposure? Is ICP4 still bound to the unwound duplex?

Minor points

13. The labelling between Figure S1 'B-scrambled' to Table S1 'B-G-rich scrambled' is not consistent for the 'G-rich sequence' mentioned on Line 87

14. Figure 3A has a label missing from the top of the immunoblot

15. The IH6 antibody is not specific for G4s (Kazemier et al 2017, NAR, 45, p5913), this should be mentioned.

Reviewer #3 (Remarks to the Author):

This study shows that G-quadruplexes (G4) bind to the HSV DNA binding protein, ICP4. Since ICP4 is known to both activate and repress transcription the authors set out connect the G4 binding capability to activation. This is an interesting idea, however there are numerous oversights with respect to the interpretation of the data. The issues are listed below:

1. Pulldowns were done with a number of oligonucleotide baits. Not surprisingly they got most of the suspected subjects based on past work. They settle on ICP4 because it was the one they most consistent got back. There was no consideration of specific ICP4 binding sites represented in the baits, the length of the baits, or the fact that ICP4 is a difficult protein to work with and prone to precipitation. Therefore, these experiments are not conclusive.

2. The experiment in fig.2 is not well controlled. There is no mock infected cells. Furthermore, the signal for the un2 probe should be lighting up entire replication compartments. It has homology to the HSV genome and at this time post infection (8h) there are huge replication compartments containing viral DNA, polIII and ICP4 in the nucleus. They only saw 1 or 2 PLA dots per nucleus and say that this , "perfectly fits with the evidence that just the single parental genome is transcribed in each cell where ICP4 acts as transcription factor," and give some references. This is not true. The authors of the cited papers were talking about input genomes, before DNA replication. This experiment was performed long after the onset of DNA replication (8h), when ICP4 and polIII are found on many genomes in the replication compartments. These results are not consistent with the a role in the transcription of the genome. What happens at 2h, which would be before DNA replication? Again a mock infection would be important.

3. The western blots in figure 3 are not of high quality. Why are there 2 broadly spaced bands for ICP4?

4. In figure 4 they show that ICP4 activated its own promoter driving the expression of a reporter gene and the tk promoter was unresponsive to ICP4. This is odd. ICP4 represses its own promoter and activates the tk promoter.
5. There should be more consideration of the presence of ICP4 binding sites in these experiments.
6. It would be helpful to provide an annotated map of all the sequences used in these experiments. They give nucleotide numbers for the specific strain used, but all strains are different. The reader shouldn't have to sort this out.

We are grateful to the reviewers for their useful comments that helped us improve our manuscript. Our answers to all of their comments are shown in red below.

Reviewers' comments:

Reviewer #1 (Remarks to the Author):

Frasson et al reported ICP4, the major HSV-1 transcription factor binds to and unfolds several viral G4s, and ICP4, through G4 binding, regulates its own promoter.

However, the work remains preliminary as definitive demonstration of a preference for a specific G4 topology is not demonstrated. In addition, the data supporting the unfolding of G4s is not convincing as currently presented.

1. The analysis performed in Figure 1 does not include a G4 of antiparallel topology, only parallel and mixed topology. Inclusion of a G4 of anti-parallel topology would strengthen their claim in line 179. Why was hTel-T (antiparallel from the CD presented in Fig S3A not used in this binding assay?). Was an i-motif tested – un2 complementary?

Before performing any pull-down assay, which requires biotinylation of the oligonucleotides, we always checked if and how the oligonucleotide conformation changed by CD. The antiparallel conformation of un2 modified to hybrid when biotin was added at the 5'-end (the same occurs when it is added at the 3'-end, not shown) (compare Fig S1, S3A and Fig S5A).

Thus, we performed CD analysis of the G4 sequences reported and characterized by antiparallel topology, i.e. gp054 from HSV-1 genome (doi: 10.1016/j.antiviral.2015.03.016), human telomeric sequence 201D (doi: 10.1002/anie.201709184) and hTel (doi: 10.1093/nar/gkw968). All these sequences, when biotinylated, did not maintain the antiparallel topology, shifting to hybrid or mixed structures. We then tested the BOM17 sequence (doi: 10.1093/nar/gku111) and found that this is the only sequence that retains the pure antiparallel topology when biotinylated: thus, this sequence was used in pull-down analysis. As gp054a maintained a fair 290 nm positive peak and it represents an HSV-1 sequence composed of 4G traits, as un2, gp054a was also included. In addition, a C-rich region (LTR-IIIc, doi: 10.1093/nar/gkz93) and the ICP4 consensus binding sequence IE3 were also added as requested. These new data have been added in the main text at lines 128-138.

Where EMSAs also performed?

We performed EMSA to check ICP4 binding to the main tested sequence, i.e. un2. Bands corresponding to ICP4-un2 complexes were obtained only with the folded oligonucleotide and not with the scrambled control. ICP4 has been previously reported to form oligomers and thus multiple complex bands in EMSA are visible (doi: 10.1128/JVI.01054-07). These data have been added in the main text at lines 163-166 and in Fig. S4D.

2. The graph in Figure 1B does not add to the immunoblot presented. The control used is an input control, a percentage of the nuclear extract added before each pulldown. An oligonucleotide that does not bind ICP4 would be more appropriate as a negative control, and then relative binding could be graphed if desired.

Binding of the test oligonucleotides has now been calculated in reference to the ICP4 consensus binding sequence IE3. Our results confirm ICP4 clear-cut preference for parallel G4s and no binding to antiparallel G4s or C-rich sequences. These aspects have been added in the main text (lines 133-136), Figure 1A and B have been modified (lines 144-151).

3. ICP4 used in the FRET unfolding assays was from IP and ultrafiltration, and its purity determined by mass spectrometry and ICP4 immunoblot. Were these samples separated by SDS PAGE and stained for total protein?

To answer to the reviewer request, ICP4 was purified with the two reported methods, i.e. immunoprecipitation (ipICP4) or ultrafiltration (ufICP4), starting from the same amount of nuclear proteins (20 µg) extracted from U2-OS infected cells (6 hpi, MOI of 2) (doi: 10.1128/JVI.00385-11; doi: 10.1128/JVI.71.2.1547-1557.1997; PMID: 8551564). The protein aliquots obtained from both procedures were subjected to buffer exchange via Amicon® Ultra and maintained in phosphate buffer (20 mM, pH 7.4) and 100 mM KCl. Each purified ICP4 aliquot was used to prepare two samples that were run on the same SDS-PAGE gel. The polyacrylamide gel was subsequently divided into two parts: one was fixed in 25% MeOH and 10% HOAc and stained in Coomassie solution (10% acetic acid in water, containing 60 mg/L of Coomassie Blue R-250), whereas the second part was transferred to a nitrocellulose membrane and analyzed by Western blot. UfICP4 was also subjected to whole bands digestion and LC-MS/MS analysis. The immunoprecipitation procedure led to purification of a tiny amount of ICP4 monomer (175 KDa) visible in the SDS gel and on the WB membrane, whereas the ultrafiltration procedure consented to obtain larger amounts of protein. As reported in the literature (10.1371/journal.pone.0078242; 10.1371/journal.pone.0070889.g001; 10.1128/JVI.00385-11; 10.1128/JVI.66.11.6398-6407.1992) ICP4 is normally purified in its multiband native state, encompassing the homodimer form (heavier than 245 KDa), the monomeric form (175 KDa) and the differentially phosphorylated bands (PMID: 8551564; 10.1371/journal.pone.0078242). In fact, in the ufICP4 Coomassie stained gel we retrieved several bands. Because ICP4 was often reported to co-precipitate with TBP (TATA BOX binding protein) and TFIID (0.1128/JVI.00385-11),

we used an extraction buffer (CellLytic™ NuCLEAR™ Extraction Kit, Merck Life Science) containing high concentrations of both DTT (1 mM) and NaCl in (500 mM) to dislodge protein complexes. We also specifically looked for the presence of TBP and TFIID in mass spectrometry analysis of uflCP4, retrieved no matching, thus reinforcing our data on protein purity.

In the case of ipICP4, the use of the anti-ICP4 antibody allowed the immunoprecipitation of the monomeric ICP4 (135 KDa) towards which the antibody has the highest affinity.

To note that the immunoprecipitation procedure has an average cost of € 150 (immune precipitation kit reagents, 30 µg of anti-ICP4 antibody, and Amicon® Ultra column), whereas the ultrafiltration procedure costs less than € 10, being more rapid and retrieving higher protein yields.

These aspects have been added in the text (lines 159-170) and the outcomes of ipICP4 and uflCP4 purification in Fig. S4A-C.

The source of the protein influenced the results presented in Table S3 (for un2 it was at least 3-fold). While mentioned, using bacterially expressed and purified ICP4 would remove these problems and the influence of other proteins.

Bacterial purification remains a critical procedure for many viral proteins, with ICP4 being one of them. The protein that is normally synthesized by the infected host cell undergoes ADP-ribosylation and phosphorylation. The accurate phosphorylation of a specific domain formed by adjacent serine residues represents a key regulatory step in ICP4 (10.1128/JVI.00385-11). Phosphorylation is carried out by PKA and PKC, at residues 175 to 198 (PMID: 8551564). Up to date, the only work that obtained an ICP4 domain purified in bacteria is doi: 10.1093/nar/gkx419, where the DNA binding domain (from aa 258 to 487) was expressed in E. coli cells. This DNA binding domain does not undergo critical post translational modifications.

Due to the fact that both C- and N- terminal are both strictly required to consent ICP4-mediated viral gene expression regulation (10.1128/JVI.00651-12), we decided to work exclusively with the protein purified from infected cells.

4. The presentation of the FRET unfolding assays in Figure 1C and S3B was confusing and a diagram or clearer explanation of the experimental procedure should be provided.

We added two bar graphs as panels F and G in figure S5 to facilitate the more immediate comprehension of the FRET data.

Line 156/157 'In the case of un2 and myc, addition of ipICP4 induced unfolding, the degree of which largely or mildly increased when the G4 complementary strand was added before or concurrently to ipICP4, respectively (Table S3 and Figure 1C)'. Does Figure 1C have the complementary strand present?

Figure S3B shows the unfolding assay when complementary strand was added at the same time as ICP4.

We now cited Fig S3B instead of Figure 1C in the text.

Why was the FRET unfolding assay not done at 37°C? Does it require ATP?

ICP4 does not require ATP or any other co-factors. FRET experiments have now been performed also at 37°C: results are summarized, as mean values of four independent experiments, in Figure S5H and lines 188-190 of the manuscript. These results are in line with the ones obtained with the fluorescence melting assay, with ICP4 destabilizing un2 and myc G4s, while causing no fluorescence variation in LTR-III and hTel sequences.

These findings would be strengthened by demonstrating the unwinding by another method such as the in vitro unwinding experiments performed with native gels as shown in Fig 2F and H of Ribeyre, et al, The Yeast Pif1 Helicase Prevents Genomic Instability Caused by G-Quadruplex-Forming CEB1 Sequences In Vivo. Plos Genetics 2009, 5 (5), e1000475. doi:10.1371/journal.pgen.1000475.

To confirm ICP4 unfolding capacity, we visualized formation of the un2 duplex at 15, 60 and 120 min upon incubation of the G4-folded un2 with its complementary strand in the absence and presence of uflCP4 by EMSA.

The presence of uflCP4 significantly increased duplex formation over samples where uflCP4 was absent. These results have been added in the text (lines 208-217) and in Fig. S7.

5. In addition, the FRET unfolding assays in Figure 1C do not to reflect the results from the CD (Figure 1D).

The ICP4-mediated unfolding ratios obtained in FRET and CD are actually are very similar: please compare FRET analysis in Figure S5F (quantification of data presented in Fig 1C) to CD data in Fig 1D

Is the small decrease in the 260nm peak observed in the CD presented in Fig 1D and 3B, just dilution from the addition of protein?

To avoid bias from dilution, we normally treat all samples with the exact same buffer: thus, in CD and FRET analysis the same amount of phosphate buffer containing 100 mM KCl or protein elution buffer was added to the non-treated oligo in parallel to the samples incubated with uflCP4 or ipICP4, respectively.

In the case of figure 1D, the buffer of the 10x ICP4 concentration was used in the control sample. In addition, note that the maximum amount of protein/buffer added was 5 µl in 300 µl of folded oligo, thus representing at the most 1.6% of the total volume.

6. The use of the proximity ligation assay to visualise a signal G4 bound to ICP4 will be of interest to others in the field. Was an immuno-FISH performed with un2 and anti-ICP4 also?

We performed the immuno-FISH experiment to detect how un2 and ICP4 were distributed in infected cells at 2-4-6-8 hpi. These data have been added in the supplementary material (Figure S9B) and commented them in the text (lines 225-233).

7 What is the sequence of the IE-CS used in Figure 3A (as it is not listed in Table S1) in the pull down.

ICP4 consensus binding sequence is now indicated as B-IE3 in Table S1.

ICP4 binds as strongly to IE-CS as some of the parallel promoter G4s. Is it a double stranded DNA consensus sequence for ICP4?

Yes, it is, indeed. ICP4 IE-consensus sequence is a duplex DNA sequence (CCGATCGTCCACACGGAGC and reverse-complement GCTCCGTGTGGACGATCGG, Table S1) present in the ICP4 promoter (doi: 10.1093/nar/gkx419). This is in line with the reported ICP4 consensus DNA duplex sequence ATCGTCNNNNYCGRC, where R is purine, Y is pyrimidine, and N is any base (doi: 10.1128/JVI.00385-11).

The ability of ICP4 to bind the G4 does not reflect the unwinding observed in Figure 3.

As reported in Table S1, the biotinylated oligos used in the cross-linking procedure have higher melting temperatures than the ones used in CD analysis. We deem that the different behavior, mainly observed on the ICP4 146947 sequence, derives from the lower T_m value of the non-biotinylated oligo, consenting increased ICP4 unfolding activity on that sequence.

Also, these are all parallel G4s, with 146574-8 (Fig S7) showing some mixed topology.

We have previously shown that HSV-1 G4 sequences fold into parallel G4s, with just a few showing a hybrid/mixed G4 structure (10.3390/molecules24132375). In HSV-1 we did not observe any antiparallel G4s.

In our manuscript we demonstrate that ICP4 binds with the highest affinity to G4s folded into the parallel conformation, but in the cross-linking experiments we still see some binding to the mixed G4s (that also contain some degree of parallel structures). The HSV-1 ICP4 sequence 146574-78 is characterized by a spectrum where the parallel structure is predominant, justifying ICP4 effect on that sequence.

8. A schematic diagram of the ICP4 promoter showing the location of each of the four G4s would assist with the interpretation of the reporter assays in Figure 4 and 5. The statement in line 216, indicates some of the G4s are downstream of the transcription start site?

The requested diagram has been added as Fig 3A.

Inclusion of a ICP4 promoter reporter construct with a mutant G4 binding sequence that could not form a G4 (ie of 14666), would strengthen the claim that ICP4 G4 unfolding is required for ICP4 activation of the promoter.

We could not produce a useful mutant sequence because mutating just few G bases would not be sufficient to remove the possibility of G4 to form in this sequence, since there are multiple G4 regions. Thus, we produced a deleted promoter region fused to the luciferase reporter gene, lacking the G-rich region from nucleotide 146629 to 146778 (i.e. 146666 G4 sequence). However, this deleted promoter showed a remarkably reduced activity, as compared to the full-length promoter and thus we deemed it not suitable for further experiments. Results of two independent luciferase experiments in U2-OS cells are shown in the graph added as Fig S11 and described in the main text at lines 295-300.

9. Figure 5C shows very consistent expression of the ICP4-EYFP plasmid across the concentrations of B19 used. Given the plasmid was transiently transfected into the U-2 OS cells, how was the usual variation in expression between transfected cells overcome in collecting this data?

The quantification was performed quantifying the fluorescence intensity of U2-OS cells transfected with ICP4-YFP and observed at 40x magnification. In total we considered 60 different fluorescent nuclei per condition from two independent experiments. For each condition we measured the fluorescence of each nucleus and calculated the mean value. The mean value was then graphed.

A wider field of cells or lower magnification should be shown to be more representative.

We have now provided a wider field with lower magnification in Fig S12.

10. The immunoblot presented in Figure 5C is not convincing given the uneven bands of tubulin. Also, was the same amount of total protein loaded as the intensity of the variation in the bands has a major influence on the interpretation of the level of ICP4 expression in response to B19 concentration.

We have now added a new image in Fig 5D that shows more even bands of tubulin.

11. The statement on line 333 is incorrect, one example of a protein binding to G4s in a topology specific manner is described in reference 11.

The statement has been amended as suggested.

12. Given factors such as ICP4 are proposed to bind and unfold G4s within promoter regions to bring in other factors and the PIC. What is the role of the unwinding? From their location is it site exposure? Is ICP4 still bound to the unwound duplex?

Considering that ICP4 has the ability to bind duplex DNA, it is likely that it remains bound to the unwound duplex where it recruits transcription factors. We hypothesize that the unwinding is necessary for the subsequent activity of the recruited transcription factors and polymerase complexes, as reported in the discussion (lines 371-376).

Minor points

13. The labelling between Figure S1 'B-scrambled' to Table S1 'B-G-rich scrambled' is not consistent for the 'G-rich sequence' mentioned on Line 87

The oligonucleotide has been named "B-G-rich scrambled" in both Table S1 and Figure S1.

14. Figure 3A has a label missing from the top of the immunoblot

Figure 3A (now 3B) has been amended.

15. The IH6 antibody is not specific for G4s (Kazemier et al 2017, NAR, 45, p5913), this should be mentioned.

We have added the reported reference and also a line stating that anti-G4 antibodies, including BG4 (10.1021/acscchembio.9b00934), while being valuable tools to investigate G4 formation in cells, they all show some degree of unspecificity (lines 754-755).

Reviewer #3 (Remarks to the Author):

This study shows that G-quadruplexes (G4) bind to the HSV DNA binding protein, ICP4. Since ICP4 is known to both activate and repress transcription the authors set out connect the G4 binding capability to activation. This is an interesting idea, however there are numerous oversights with respect to the interpretation of the data. The issues are listed below:

1. Pull-downs were done with a number of oligonucleotide baits. Not surprisingly they got most of the suspected subjects based on past work.

The pull-down assays coupled to mass spectrometry had been previously successfully employed by our group to identify cellular G4 interactors in the case of other important pathogens, such as HIV-1. In any case, to further prove the soundness of our pull-down results, we tested ICP8, one of the major HSV-1 proteins involved in viral DNA replication, that was not detected in our pull-down assay, while being previously reported to partially overlap with the anti-G4 antibody signal in infected cells. We both purified ICP8 from bacterial cells or used the viral ICP8 from infected cells and challenged its ability to bind different viral and cellular G4 sequences. We used different techniques (EMSA, cross-linking experiments) but we never recorded any ICP8-G4 binding, confirming the reliability of our pull-down data. These new data have been added in the text (lines 105-121) and in Figure S2.

They settle on ICP4 because it was the one they most consistent got back. There was no consideration of specific ICP4 binding sites represented in the baits, the length of the baits, or the fact that ICP4 is a difficult protein to work with and prone to precipitation. Therefore, these experiments are not conclusive.

We obviously considered if reported ICP4 consensus sequences were present in our baits. As mentioned in the text, they were not present in any of them.

ICP4 was chosen as G4s have been characterized as important regulators at promoters in both human cells and viruses. The choice to focus on ICP4 was based on this knowledge and the fact that such an important viral transcription factor was quite appealing to try and shed additional light on HSV-1 biology. In addition, ICP4 activity at promoters has not been unambiguously understood yet. Thus, even if we were aware of the difficulties to be faced working with this protein, we chose the harshest but also most interesting path for our research.

2. The experiment in fig.2 is not well controlled. There is no mock infected cells.

Mock infected cells and control samples were present in Fig S6B (now Fig S9C)

Furthermore, the signal for the un2 probe should be lighting up entire replication compartments. It has homology to the HSV genome and at this time post infection (8h) there are huge replication compartments containing viral DNA, polIII and ICP4 in the nucleus. They only saw 1 or 2 PLA dots per nucleus and say that this , “perfectly fits with the evidence that just the single parental genome is transcribed in each cell where ICP4 acts as transcription factor,” and give some references. This is not true. The authors of the cited papers were talking about input genomes, before DNA replication. This experiment was performed long after the onset of DNA replication (8h), when ICP4 and polIII are found on many genomes in the replication compartments. These results are not consistent with the a role in the transcription of the genome. What happens at 2h, which would be before DNA replication? Again a mock infection would be important.

It has been reported that ICP4 foci depend on the initial MOI (10.1128/JVI.78.4.1903-1917.2004). From the immune FISH images that we have now provided (Figure S9B and lines 225-233 in the revised manuscript), where both HSV-1 G4 un2 and ICP4 signals are present, more ICP4-un2 interactions appear to be present in the infected cell. However, the PLA is a more stringent technique, which allows to detect only interacting molecules, i.e. those that are at 40 nm or less of distance. Previous data on ICP4 have always been performed at high MOIs (usually MOI from 5 to 10) (10.1128/JVI.72.4.3307-3320.1998; 10.1128/JVI.00651-12, 10.7554/eLife.51109.001), thus forcing the number of initial parental genomes per cell at numbers higher than one to better study the ICP4-DNA relationship.

In our case, we performed PLA at the lowest possible MOI to allow visualization of the interaction in conditions as close as possible to the ones reported to mimic the natural HSV-1 infection (10.1371/journal.ppat.1002852; 10.1371/journal.ppat.1006721), and in the meanwhile obtaining a good amount of infected U2-OS cells.

3. The western blots in figure 3 are not of high quality. Why are there 2 broadly spaced bands for ICP4?

ICP4 can be detected as monomer, homodimer, and also in differently phosphorylated isoforms. The distance between the bands depends on SDS-PAGE conditions (acrylamide percentage and running time). The presence of more than one band of ICP4 in SDS-PAGE was also reported by others (10.1371/journal.pone.0078242; 10.1371/journal.pone.0078242.; 10.1128/JVI.01869-06; 10.1128/jvi.01869-06; PMID: 8551564).

4. In figure 4 they show that ICP4 activated its own promoter driving the expression of a reporter gene and the tk promoter was unresponsive to ICP4. This is odd. ICP4 represses its own promoter and activates the tk promoter.

It has been reported that crude extracts from HSV-1 infected cells containing ICP4 led to TK promoter activation (10.1128/JVI.78.12.6162-6170.2004; 10.1128/JVI.78.12.6162-6170.2004). However, ICP4 complexes with DNA sequences representing TK promoter flanking regions have also been reported (PMID: 2159535).

To prove that the TK promoter was activated by other viral actors, we measured TK-luciferase activity in a plasmid transfected into cell that were subsequently infected with HSV-1, in addition to cells that overexpressed ICP4. We proved that the TK promoter was activated upon HSV-1 infection in a MOI-dependent manner (see figure below, left panel), while it was not activated in ICP4 expressing cells (Fig 4B). The TK promoter was also activated in CMV infected cells, in a MOI-dependent manner (see Figure below, right panel). Since CMV lacks an ICP4 homologue, we deem these data sufficient to prove that ICP4 is not the factor that stimulates TK promoter activity. Besides, TK-promoter-driven reporter plasmids are used as internal controls also in studies encompassing single CMV major transcription factors (10.1016/j.chembiol.2015.12.012), reinforcing the hypothesis that this promoter is prone to respond to a multifactorial environment strictly connected to the onset of viral infection, rather than to be activated by a single protein.

5. There should be more consideration of the presence of ICP4 binding sites in these experiments.

The reported ICP4 binding site (ATCGTCNNNNYCGRC, where R is purine, Y is pyrimidine, and N is any base, (10.1093/nar/gkx419) is a double stranded DNA. In our experiments, despite the lack of that sequence, a single-stranded DNA is folded into the four-stranded G4 structure. Our work does not question the fact that ICP4 binds to the reported consensus sequence, as also proven by our pull-down assay (Fig. 1B), but it shows that parallel G4s structures can be bound as well, and even at a higher extent than the reported consensus sequence (Fig. 1B). We thus characterized another type of binding between ICP4 and DNA.

6. It would be helpful to provide an annotated map of all the sequences used in these experiments. They give nucleotide numbers for the specific strain used, but all strains are different. The reader shouldn't have to sort this out.

We have added a scheme showing where the G4 forming sequences in the ICP4 promoter are located. (Fig. 3A).

REVIEWERS' COMMENTS:

Reviewer #1 (Remarks to the Author):

The authors have addressed the my concerns in a satisfactory manner, providing a detailed response with appropriate changes where necessary to the manuscript.